# Antibacterial Activity and Antifungal Activity of Monomeric Alkaloids

**DOI:** 10.3390/toxins16110489

**Published:** 2024-11-12

**Authors:** Amin Mahmood Thawabteh, Aseel Wasel Ghanem, Sara AbuMadi, Dania Thaher, Weam Jaghama, Rafik Karaman, Laura Scrano, Sabino A. Bufo

**Affiliations:** 1Department of Chemistry, Birzeit University, Birzeit P.O. Box 14, Palestine; athawabtah@birzeit.edu; 2Faculty of Pharmacy, Nursing and Health Professions, Birzeit University, Birzeit P.O. Box 14, Palestine; aseel.wasel@gmail.com (A.W.G.); abumadisarah@gmail.com (S.A.); daniathaher2000@gmail.com (D.T.); weam_rezeq@yahoo.com (W.J.); 3Pharmaceutical Sciences Department, Faculty of Pharmacy, Al-Quds University, Jerusalem 20002, Palestine; dr_karaman@yahoo.com; 4Department of Sciences, University of Basilicata, Via dell’Ateneo Lucano 10, 85100 Potenza, Italy; 5Department of European and Mediterranean Cultures, University of Basilicata, Via Lanera 20, 75100 Matera, Italy; laura.scrano@unibas.it; 6Department of Geography, Environmental Management and Energy Studies, University of Johannesburg, Auckland Park Kingsway Campus, Johannesburg 2092, South Africa

**Keywords:** alkaloids, antibacterial, antifungal, pathogenic bacteria, Gram-positive, Gram-negative, pathogenic fungi, filamentous fungi, minimum inhibitory concentration (MIC), medicinal effect

## Abstract

Scientists are becoming alarmed by the rise in drug-resistant bacterial and fungal strains, which makes it more costly, time-consuming, and difficult to create new antimicrobials from unique chemical entities. Chemicals with pharmacological qualities, such as antibacterial and antifungal elements, can be found in plants. Alkaloids are a class of chemical compounds found in nature that mostly consist of basic nitrogen atoms. Biomedical science relies heavily on alkaloid compounds. Based on 241 papers published in peer-reviewed scientific publications within the last ten years (2014–2024), we examined 248 natural or synthesized monomeric alkaloids that have antifungal and antibacterial activity against Gram-positive and Gram-negative microorganisms. Based on their chemical structure, the chosen alkaloids were divided into four groups: polyamine alkaloids, alkaloids with nitrogen in the side chain, alkaloids with nitrogen heterocycles, and pseudoalkaloids. With MIC values of less than 1 µg/mL, compounds **91**, **124**, **125**, **136**–**138**, **163**, **164**, **191, 193**, **195**, **205** and **206** shown strong antibacterial activity. However, with MIC values of below 1 µg/mL, compounds **124**, **125**, **163**, **164**, **207**, and **224** demonstrated strong antifungal activity. Given the rise in antibiotic resistance, these alkaloids are highly significant in regard to their potential to create novel antimicrobial drugs.

## 1. Introduction

Because of their wide range of biological actions, alkaloids stand out among the rich, diverse spectrum of natural chemicals that scientists are exploring in the constantly changing field of pharmacology in search of new and effective antimicrobial agents [1,2,3]. Due to their strong pharmacological characteristics, these naturally occurring substances, found in a variety of plants, have long been utilized in medicine [4,5,6].

The urgent need for novel treatments is highlighted by the rising incidence of antibiotic-resistant bacteria, such as methicillin-resistant *Staphylococcus aureus* (MRSA) [7,8,9]. Because of their distinct structures and modes of action, alkaloids present a viable way toward the development of novel antimicrobial drugs that can fight these resistant pathogens and lessen the prevalence of infectious disorders worldwide [10,11,12,13].

This article describes the many kinds of alkaloids, such as isoquinoline, indole, pyridine, and steroidal alkaloids, as well as their modes of action against microorganisms based on randomly selected published papers in peer-reviewed scientific publications within the last ten years. It also examines the role and effectiveness of alkaloids in antimicrobial activity. It emphasizes how crucial it is to ascertain the minimum inhibitory concentration (MIC) of these substances in order to assess their potential as antimicrobial agents. The review also looks at how alkaloids affect strains that are resistant to several drugs, providing information on how these natural substances might be used to treat some of the most difficult problems in infectious disease therapy today. This review seeks to promote pharmacological approaches to infection management by providing insight into the antibacterial potential of alkaloids in pathogen control through a thorough investigation [9,10,11,12].

## 2. Historical Use of Alkaloids as Antimicrobials

The structure and function of alkaloids, which are heterocyclic nitrogen molecules, differ greatly. This structural variety depends on their capacity to interact with biologically significant molecules, including proteins, enzymes, and receptors. Alkaloids’ biological activity depends on the formation of hydrogen bonds, which are made possible by a proton-accepting nitrogen atom and one or more proton-donating amine hydrogen atoms [12,13].

Alkaloids are a broad class of naturally occurring substances distinguished by the presence of basic nitrogen atoms. More than 18,000 distinct alkaloids have been found, and they are present in more than 300 plant families, bacteria, fungi, and mammals. These substances are well known for their numerous pharmacological actions, such as antibacterial, anticancer, antiviral, and central nervous depressant effects [12,13,14]. Alkaloids’ strong biological properties have historically made them important in conventional and contemporary medicine. It is well established that they can be used to treat infections, especially those that show signs of multidrug resistance (MDR). Bacteria, fungi, and viruses are among the pathogenic microorganisms against which alkaloids like berberine, sanguinarine, and piperine have shown potent antibacterial activity [12,15,16].

Alkaloids’ antibacterial properties are mainly ascribed to their capacity to disrupt cell membranes, suppress enzyme activity, and obstruct target microbes’ production of proteins, RNA, and DNA. Some alkaloids are effective against organisms that are resistant to traditional antibacterial treatments because they help boost the immune system and prevent the formation of biofilms [12,13,14,15,16]. Alkaloids’ varied structures and modes of action, which allow them to efficiently target a wide variety of pathogenic microbes, provide strong evidence for their participation in antimicrobial activity. Their history and current use in pharmacology highlight their potential as useful agents in the fight against infectious diseases, including those that are resistant to conventional therapies [10,11,12,13,14,15,16].

## 3. Alkaloids Classification

Alkaloids can be categorized based on their chemical structure, biosynthesis origin, or pharmacological activity. A general classification and Figure 1 and Figure 2 are provided below. Dimer and monomeric alkaloids are another classification that is used [16,17]. These dimers can be produced in a lab or found naturally [18,19]. True alkaloids, protoalkaloids, polyamine alkaloids, peptide alkaloids, and pseudoalkaloids are among the several categories of monomeric alkaloids that are generated from amino acids [17,18].

Alkaloids that include nitrogen heterocycles are known as true alkaloids. These include tropane alkaloids, such as atropine and scopolamine; pyridine alkaloids, such as nicotine; isoquinoline alkaloids, such as morphine; and indole alkaloids, such as serpentine and reserpine. β-Phenylethylamine, colchicine, muscarine, and benzylamine derivatives are examples of proto-alkaloids, which are alkaloids containing nitrogen in the side chain [19,20,21,22,23].

Compounds with more than one amine group are called polyamines. Putrescine, spermidine, and spermine are the three most well-known polyamines. All live cells contain them and play a role in many cellular functions. Because of their unique structural characteristics and peptide bonds with **13**-, **14**-, and **15**-membered cycles, peptide alkaloids may be identified from other alkaloids. Terpenes, steroids, and other alkaloids with a basic carbon skeleton not derived from an amino acid are known as pseudoalkaloids [19,20,21,22,23].

## 4. Mechanisms of Antimicrobial Action of Alkaloids

Alkaloids utilize their antibacterial properties via a variety of methods. Among the main mechanisms of action are:Inhibition of Nucleic Acid and Protein Synthesis

A key component of alkaloids’ antibacterial activity is their substantial capacity to interfere with the creation of proteins and nucleic acids inside bacterial cells. Chelerythrine, for example, has been shown to prevent nucleic acid synthesis and cellular division in several species, including MRSA [24]. Berberine functions as an efficient DNA intercalator, interfering with DNA replication, RNA transcription, and protein production [25,26]. Because of these interactions, DNA and RNA undergo structural alterations that make it impossible to function as normal templates for vital biological functions [26].

Effects on Bacterial Cell Membrane Permeability

Another important antibacterial action of alkaloids is their capacity to change the permeability of bacterial cell membranes. Because of their high lipophilicity, alkaloids like 8-hydroxyquinoline can pass through bacterial cell membranes, reach their target locations, and have antibacterial effects [27]. In a similar vein, the PA-1 cell line’s increased membrane permeability causes a notable uptake of crystal violet, indicating serious membrane damage, which contributes to its bactericidal activity [28].

Inhibition of Efflux Pumps

A number of alkaloids are potent efflux pump inhibitors (EPIs), which are essential in the battle against bacterial resistance. For instance, piperine inhibits the NorA efflux pump in S. aureus, which increases the accumulation of medicines like ciprofloxacin [29]. Higher intracellular concentrations of antibiotics are made possible by this inhibition, which increases the antibacterial efficacy of the drugs. By preventing the efflux of antibiotics from bacterial cells, the alkaloid resperpine is also known to reverse multidrug resistance [30,31].

Interference with Metabolic Pathways

Another crucial aspect of alkaloids’ antibacterial activity is their ability to disrupt the metabolic processes of bacteria. For example, the activity of enzymes essential to bacterial viability is inhibited by Michellamine B [32]. The growth and proliferation of dangerous bacteria (*Staphylococcus aureus*, *Bacillus subtilis*, *Yersinia enterocolitica*, *Escherichia coli*, *Klebsiella pneumoniae*, and *Candida albicans*) are considerably inhibited by this disruption of enzymatic processes, which results in a poor bacterial metabolism [32].

## 5. Monomeric Alkaloids

### 5.1. Alkaloids with Nitrogen Heterocycles (True Alkaloids)

These compounds are made from cyclic amino acids, such as ornithine, tyrosine, phenylalanine, lysine, histidine, tryptophan, arginine, glycine, and aspartic acid, and they contain intracyclic nitrogen. They are complex, biologically active substances. Lactic acid, malic acid, tartaric acid, acetic acid, and citric acid are the organic acids in the natural salts that create them [33,34,35].

#### 5.1.1. Pyrrolidine Alkaloids

Four carbon atoms and one nitrogen atom make up the five-membered ring of the saturated heteromonocyclic molecule pyrrolidine (**1** in Figure 3). In vitro tests were conducted to evaluate some pyrrolidine derivatives’ antibacterial and antifungal properties [36,37]. Sodium pyrrolidide (2,6-dipyrrolidino-1,4-dibromobenzene; 2,4,6-tripyrrolidino chlorobenzene; and 1,3-dipyrrolidino benzene) and 1,2,3,5-tetrahalogeno benzenes were present during the synthesis of these derivatives. Numerous harmful bacteria were hindered in their growth by the resultant compounds, including 2,6-dipiperidino-1,4-dibromobenzene and 2,4,6-tripyrrolidinochlorobenzene (**2** and **3** in Figure 3). No antibacterial activity was shown by 1,3-dipyrrolidinobenzene when evaluated in the same study. Thus, we may conclude that the halogen substituents are primarily responsible for the bioactivities of molecules **2** and **3**. In this case, the combination of the bromo-, chloro-, and pyrrolidine substituents demonstrated inhibitory efficacy against the microbes [38].

The spiro pyrrolidines 4a–c (4–6 in Figure 3) were created by the 1,3-dipolar cycloaddition reaction of chalcones with azomethine ylide (4a–c). Compound c has antibacterial activity against Gram-positive bacteria, such as *B. subtilis* and *Enterococcus faecalis*, with MIC values of 75 and 125 µg/mL, and against Gram-negative bacteria, such as *E. coli* and *Pseudomonas aeruginosa*, with MIC values of <125 and 150 µg/mL. In contrast, compounds a and b show antibacterial activity with MIC values of >100 µg/mL against both *B. subtilis* and *E. faecalis* and >125 µg/mL MIC values against *E. coli* and *P. aeruginosa* [39,40]. Table 1 lists more pyrrolidine alkaloids along with their activities.

#### 5.1.2. Tropane Derivatives

The *N*-methyl group that joins the piperidine and pyrrolidine rings is what distinguishes tropane (**25** in Figure 4). Because of their wide range of biological activity and structural adaptability, tropane derivatives have become increasingly popular in antimicrobial research [60,61]. These alkaloids have strong antibacterial qualities, offering a promising method for creating new therapeutic medicines to combat various infections [61,62,63]. *Atropa belladonna* is the source of the anticholinergic medication atropine (**26** in Figure 4). The mixture of d- and l-hyoscyamine is racemic. It has a strong antiviral effect and has been evaluated for antibacterial properties. With an MIC of 1000 ppm, it also prevented B. ceruse from growing [64,65,66].

The compound 3-(3′-methoxytropoyloxy)-6-tigloyloxy-7-hydroxy tropane (**27** in Figure 4) is a tropane derivative extracted from *Datura stramonium* (Solanaceae). Compound **27** extract has exhibited antibacterial activity against Gram-negative bacteria, and *K. Pneumonia*, *P. aeruginosa*, *Shigella boydii*, and *Salmonella typhi* have the lowest amount of observed activity against *E. coli*. Among the Gram-positive bacteria, it was only active against *S. aureus* [67].

Hyoscyamine (**28** in Figure 4) is an antimuscarinic agent extracted from *Solanaceae*. It is the levo-isomer of atropine [68,69]. Scopolamine (**29** in Figure 4) is an antimuscarinic agent extracted from *Solanaceae*. It is the (S)-tropic acid ester of 6beta,7beta-epoxy-1alphaH,5alphaH-tropan-3alpha-ol. Its isolate from *Datura stramonium* demonstrated activity against bacteria and fungi, specifically against the *B. subtilis*, *S. aureus*, and *E. faecali* bacteria and fungi like *Aspergillus niger*, *Trichophyton rubrum*, and *Aspergillus flavus*. This isolate was analyzed and showed the presence of hyoscyamine and scopolamine [70,71,72]. Other tropane derivatives are shown in Table 1.

#### 5.1.3. Pyrrolizidine Derivatives

Pyrrolizidine (**30** in Figure 4) derivatives comprise a necine base, a double five-membered ring with a nitrogen atom in the middle, and one or two carboxylic esters called necic acids [73,74]. These alkaloids are secondary metabolites that are synthesized by plants primarily as a defense mechanism against herbivores, insects, and pathogens. They have been found in the plant families Asteraceae, Boraginaceae, Fabaceae, and Orchidaceae [73,74,75].

Several antimicrobial activities of pyrrolizidine derivatives have been identified as having mild to strong effects against bacteria such as *E. coli* and *P. Chrysogenum*. In particular, Lasiocarpine and 7-angeloyl heliotrine (**31** and **32** in Figure 4) were observed to have significant activity against these microbes. These derivatives have been discovered to induce cell death in these bacteria by attacking bacterial cell membranes. Retronecine (**15** in Figure 3) has been found to slow the growth rate of several strains of the fungus *Fusarium oxysporum* [75,76].

A novel synthesized pyrrolizidine alkaloid (PA-1) (**33** in Figure 4) was tested and showed antibacterial solid activity with MIC values ranging from 0.0039 to 0.025 mg/mL. It was most active against *S. aureus* and *E. coli.* PA-1 exhibited the complete death of *S. aureus* and *E. coli* within 8 h [74,75].

An extract from the propolis of *Scaptotrigona aff. postica* was tested for its antimicrobial activity and was found to be active against Gram-positive and Gram-negative bacteria. Many pyrrolizidine derivatives were found in this extract, like lithosenine, 7-angeloyl-9-(2,3-dihydroxybutyryl) retronecine, 7-(2-methylbutyryl) retronecine, and 7-seneciol-9-sarracinoyl-retronecine (**34**–**38** in Figure 4). Table 1 shows other pyrrolizidine derivatives that have antimicrobial activity [75,76,77].

#### 5.1.4. Piperidine Derivatives

The structure of piperidine (**39** in Figure 4) and piperidine derivatives, which are L-lysine derivatives, consists of one amine, five methylene groups, and a ring with six radicals [71,78]. This category includes alkaloids such as coniine, lobeline, cynapine, and solenopsin. Plants belonging to the Lobeliaceae family contain piperidine alkaloids; in particular, *Lobelia inflata* includes the alkaloid lobeline [78,79]. Additionally, it is present in *Punica granatum* as pelletierine and in black pepper (*Piper nigrum*) as piperine. Piperine (**40** in Figure 4) is an N-acylpiperidine that exhibits potent activity mostly against *C. albicans*, followed by *E. coli*, with less activity being shown against *P. aeruginosa*, with MIC values ranging from 3.125 to 100 mg/mL [24,78,79,80]. Some piperidine derivatives were tested in vitro for their antibacterial and antifungal activities. These derivatives were synthesized in the presence of 1,2,3,5-tetrahalogeno benzenes and sodium piperidine (2,6-dipiperidino-1,4-dihalogenobenzenes).

The resulting compounds, 2,6-dipiperidino-4-bromochlorobenzene and 2,6-dipiperidino-4-bromoiodobenzene (**41** and **42** in Figure 5), inhibited the growth of *S. aureus*, *B. subtilis*, *Y. enterocolitica*, *E. coli*, *K. pneumoniae*, and *C. albicans* at MIC values ranging from 32 to 512 μg/mL, while 2,6-dipiperidino-4-chloroiodobenzene did not show any antibacterial activity when evaluated in the same study. Thus, we may conclude that the type and the position of the halogen and piperidine tetrasubstituents had significantly different effects on the growth of microorganisms [81].

The antimicrobial activity of six piperidine derivatives (**43**–**48**, Figure 5) was investigated. Piperidine compounds with methyl or ethyl groups on C3 and replacements of pyridine (**46**, **47**, and **48**), benzaldehyde (**43** and **45**), and 4-cyano phenyl (**44**) on C2 and C6 of the piperidne ring showed modest inhibitory actions. The antibacterial activity was increased by both the electron-donating and electron-withdrawing groups that were present as replacements on the piperidine ring. In contrast to **43**–**48**, the hydroxy, methyl, and nitro substitutions on the phenyl ring of the piperidine derivative **44** demonstrated strong inhibition against the studied bacterial species, indicating enormous antibacterial potential.

Low inhibitory action was demonstrated by compounds **45** and **46**, which the replacement of a 4-bromo group on NH-substitution on piperidine ring C4 may have caused. Compounds **43**–**48** did not show any inhibitory effect against *F. verticilliodes*, *P. digitatium*, and *C. utilis*. The addition of 4-bromo, which is joined to an amino (NH) group on C6 of the piperidne ring, may have been the cause of compounds **45** and **46**’s lack of inhibitory activity towards the fungus. Different levels of inhibition against *Candida albicans*, *S. cerevisiae*, *A. flavus*, and *A. niger* were shown by **43**, **44**, **47**, and **48**.

Compound **43** showed a moderate (5 mm) inhibitory effect *on Candida albicans*, *S. cerevisiae*, *A. niger*, and *A. flavus*. This could have resulted from the substitution of an ethyl or methyl group for the piperidine ring’s C3, which is known to boost antifungal efficacy. Additionally, **44** showed strong inhibitory action (≥5 mm) against *S. cerevisiae*, *A. niger*, *A. flavus*, and *C. albicans*. This could have arisen from the pyridine group on the C2 and C6 of the piperidine ring, which donates electrons. Compounds **43** and **44** have shown better action than compounds **45**–**48**. Similar to **47** and **48**, fluoro is a modification in the phenyl group of a piperidine ring that has been shown to have ≥5 mm inhibitory activity against *A. niger* and *C. albicans* [82]. More piperidine derivatives are shown in Table 1.

#### 5.1.5. Quinolizidine Alkaloids

Quinolizidine (**49** in Figure 5) and quinolizidine derivatives derived from the amino acid *L*-lysine have simple to complicated structural diversities [83]. They are generated by the fusion of two rings of six members that share a nitrogen atom [84]. They are primarily found in the Leguminosae family, particularly in the genera *Lupinus* and *Thermopsis*, *Baptisia*, *Genista*, *Cystus*, and *Sophora* [83,84,85,86].

Quinolizidine alkaloids extracted from *Genista sandrasica* were analyzed and studied for their antimicrobial activity. The main quinolizidine alkaloids found in the *G. sandrasica* extract were sparteine, N-acetylcytisine, 13-methoxylupanine, anagyrine, and baptifoline (**50**–**54** in Figure 4). This extract was active against *B. subtilis* and *S. aureus*, with MIC values of 31.25 and 62.5 µg/mL [87].

N-methylcytisine (**55** in Figure 5) is a natural quinolizidine compound found in *Maackia tenuifolia* and *Thermopsis lanceolata*. It showed good antibacterial activity against *E. faecalis*, with an MIC value of 20.8 μg/mL [88]. Additionally, 3-(4-hydroxyphenyl)-4-(3-methoxy-4-hydroxyphenyl)-3,4-dehydroquinolizidine, cermizine C, and jussiaeiine B (**56**–**58** in Figure 4) are quinolizidine alkaloids isolated from the *Sophora* L. plant. They were found to be active against *S. aureus*, with MIC values of 8.0, 3.5, and 6.0 g/L, respectively [89]. Jussiaeiine B also was active against *E. coli*, with an MIC value of 0.8 g/L [90]. More quinolizidine derivatives are shown in Table 1.

#### 5.1.6. Indolizidine Derivatives

The amino acid L-lysine serves as the precursor to the indolizidine alkaloid (**59** in Figure 5) and its derivatives. Indolizidine alkaloids are classified as heterocyclic alkaloids because they contain fused rings with six and five members, incorporating a nitrogen atom [91,92]. These alkaloids have been discovered in various species of the genus *Ipomoea* [92]. Most plants in this genus contain the alkaloid swainsonine, which is present in *I. carnea*. However, swainsonine was first isolated from plants of the genus *Swainsona* [93]. The primary indolizidine alkaloids are castanospermine and swainsonine [93,94].

Swainsonine (**60** in Figure 5) is an indolizidine alkaloid with three hydroxy groups at positions 1, 2, and 8. Extracts from the leaves of *Swainsona formosa* have shown antibacterial activity against *Alcaligenes faecalis*, *Aeromonas hydrophila*, *K. pneumoniae*, *Proteus mirabilis*, *B. cereus*, *S. aureus*, and *Streptococcus pyogenes*, with MIC values ranging from 150 µg/mL to less than 1000 µg/mL [95].

Castanospermine (**61** in Figure 5) is an indolizidine alkaloid with four hydroxyl groups at positions 1, 6, 7, and 8. It can be found in *Alexa canaracunensis* and *Alexa grandiflora* [87,92]. Castanospermine exhibits antiviral activity and has been shown to prevent mortality in mice infected with the dengue virus at 10, 50, and 250 mg/kg doses. Additionally, it has demonstrated activity against HIV-1 [87,92]. Celgosivir (**62** in Figure 5) is a 6-O-butanoyl castanospermine extracted from the Australian chestnut, which shows activity against the hepatitis C virus. It also exhibits activity against HSV-1, bovine viral diarrhea, and murine leukemia [96,97]. Further indolizidine alkaloid derivatives are shown in Table 1.

#### 5.1.7. Pyridine Derivatives

Pyridine (**63** in Figure 6) and its derivatives are heterocyclic compounds consisting of a ring of five carbon atoms and one nitrogen atom, and they are derivatives of the amino acid L-ornithine. Pyridine alkaloids exhibit significant antimicrobial activity against various pathogens, suggesting their potential as broad-spectrum antimicrobial agents [98,99,100].

The compound 2-cyanomethylthiopyridine-4-carbonitrile (**64** in Figure 6) is a new pyridine derivative with excellent antibacterial activity against *Mycobacterium kansasii*, with MIC values ranging from 8 to 4 μmol/L. A study on synthesizing new pyridine derivatives and their antimicrobial activity has been published [101]. Among these derivatives, two compounds, coded as 12a and 15 (**65** and **66** in Figure 6), showed good antimicrobial activity. Compound 12a exhibited activity against *E. coli*, with an MIC value of 0.0195 mg/mL, and was also active against *Bacillus mycoides* and *C. albicans*, with MIC values of 0.0048 mg/mL. Compound 15 showed activity against *E. coli*, with MIC values of 0.0048 mg/mL, and inhibited the growth of *B. mycoides* and *C. albicans*, with MIC values of 0.0098 and 0.039 mg/mL, respectively [102].

Six novel alkaloid derivatives from hybrid sulfaguanidine moieties, coded as 2a, 2b, 2d, 3a, 8, and 11 (**67**–**72** in Figure 6), demonstrated moderate to good antimicrobial activity against Gram-positive and Gram-negative bacterial and fungal strains. MIC values ranged from 4.69 to 22.9 µM against *B. subtilis*, 5.64 to 77.38 µM against *S. aureus*, 8.33 to 23.15 µM against *E. faecalis*, 2.33 to 156.47 µM against *E. coli*, 13.40 to 137.43 µM against *P. aeruginosa*, and 11.29 to 77.38 µM against *S. typhi*. They also exhibited antifungal properties against *C. albicans*, with MIC values of 16.69 to 78.23 µM, and against *Fusarium oxysporum*, with MIC values of 56.74 to 222.31 µM [103]. There were some interesting SAR study observations, such as whether the 2-cyanoacrylamide derivatives 2a,2b and 2d (**67**–**69**) exhibited good to promising activity, and the order of activity can be presented as follows: 2d (**69**) > 2a (**67**) > 2b (**68**). For the antibacterial strains, it was found that the 3-(3-hydroxy-4-methoxyphenyl)acrylamide derivative in **69** showed higher activity than the 3-(4-methoxyphenyl)acrylamide derivative in **68** and the 3-(4-chlorophenyl)acrylamide derivative in **67**. Furthermore, the 3,5-dicyano pyridine-2-one with 4-chlorophenyl in position four of pyridine derivatives 3a (**70**) showed broad and higher activity than 3-(4-chlorophenyl) acrylamide derivative 2a (**67**), and **70** was produced from the reaction of acrylamide derivative **69** with malononitrile [103]. Table 2 reports other pyridine derivatives.

#### 5.1.8. Isoquinoline Derivatives

Isoquinoline derivatives are naturally found in plant families such as Amaryllidaceae, Annonaceae, Fumariaceae, Rutaceae, Papaveraceae, and Berberidaceae, which are among the most common in the plant kingdom. These heterocyclic aromatic molecules are derived from the amino acids tyrosine and phenylalanine. Isoquinoline (**98** in Figure 7) and its derivatives are known for their significant antimicrobial activities [121,122,123,124,125].

Berberine (**99** in Figure 7) is a quaternary ammonium salt of protoberberines. It is found in many plants, such as *Hydrastis canadensis* (goldenseal), Oregon grape, barberry, and tree turmeric. While berberine is toxic when administered parenterally, it is effective when used orally against various parasitic and fungal infections. In vitro studies have shown that berberine inhibits the effects of *E. coli* and *Vibrio cholerae* enterotoxins. Berberine chloride has demonstrated activity against 43 tested strains of *S. aureus*, with MIC values ranging from 0.25 to 0.5 mg/mL, and against *C. albicans*, with MIC values of 0.1–0.2 mg/mL [126,127,128].

Thalicfoetine (**100** in Figure 7), extracted from the roots of *Thalictrum foetidum*, has a spirotetrahydropyridine-furanone core and exhibits excellent antibacterial activity against *B. subtilis*, with an MIC value of 3.12 µg/mL [129].

*Penicillium spathulatum* contains the naturally occurring chemicals spathullin A (6,7-dihydroxy-5,10-dihydropyrrolo[1,2-b]isoquinoline-3-carboxylic acid (**101** in Figure 7)) and spathullin B (5,10-dihydropyrrolo[1,2-b]isoquinoline-6,7-diol (**102** in Figure 7)). These compounds have shown effectiveness against *S. aureus*, *K. pneumoniae*, *E. cloacae*, *P. aeruginosa*, and *Acinetobacter baumannii*, with spathullin B demonstrating greater efficacy against all tested pathogens [130].

An investigation into the antibacterial efficacy of isoquinoline alkaloids isolated from *Chelidonium majus* revealed that chelerythrine (**103** in Figure 7) was the most effective against *P.* aeruginosa, with an MIC value of 1.9 µg/mL [131]. Allocryptopine (**104** in Figure 7) exhibited efficacy against *S. aureus* at 125 µg/mL; sanguinarine (**105** in Figure 7) showed more effective action against *S. aureus*, with an MIC value of 1.9 µg/m [131,132,133]; and Chelidonine (**106** in Figure 7) demonstrated antifungal efficacy against *C. albicans*, with an MIC value of 62.5 mg/L [134,135]. More isoquinoline derivatives are shown in Table 2.

#### 5.1.9. Oxazole Derivatives

Oxazole derivatives are chemical compounds with an oxazole ring, a five-membered heterocyclic ring structure comprising one nitrogen atom and one oxygen atom. The general structure of an oxazole includes an oxygen atom at position 3 and a nitrogen atom at position 1. Various substituents can be added to the oxazole ring to create a range of molecules with unique chemical and biological properties. Due to their unique structural characteristics, oxazole (**107** in Figure 7) and its derivatives may exhibit various biological activities, including antimicrobial properties [111,136,137,138].

In one investigation, researchers developed a novel family of antibacterial drugs (including 2,4-disubstituted oxazol-5-one (3a–3g)), which were prepared by reacting 1-phenyl-3-*p*-tolyl-1Hpyrazole-4-carbaldehyde with hippuric acid and sodium acetate, in an acetic anhydride. The antibacterial and antifungal properties of each tested chemical were assessed in vitro. Compounds 3a, 3c, 3d, and 3e (**108**–**111** in Figure 7) generally exhibited antibacterial activity against *E. coli*, *P. aeruginosa*, and *S. aureus* at concentrations of 25–30 µg/mL, with zones of inhibition ranging from 6.1 to 13.7 mm. These compounds also demonstrated antifungal activity at higher dosages (45–60 µg/mL) [139,140].

Various oxazole-2-amine analogs were tested for their antibacterial activity. The compounds (E)-4-(benzofuran-2-yl)-N-benzylideneoxazol-2-amine, and (E)-N-(4-nitrobenzylidene)-4-(benzofuran-2-yl)oxazol-2-amine (**112**, **113** in Figure 7) showed excellent activity against *S. aureus* and *E. coli* [141].

Nine methyl-2-(arylideneamino)oxazol-4-ylamino)benzoxazole-5-carboxylate derivatives (VIIa-h, **114**–**122** in Figure 8) have been synthesized by ethanol refluxed of 2-brmoacetylbenzofuran and urea with the presence of 5% sodium acetate. Compounds **114**–**122** have been evaluated for their antibacterial activity against both Gram-positive and Gram-negative bacteria [142,143]. The results indicate that all the compounds showed excellent antibacterial activity against *B. subtilis*, *S. aureus*, *E. coli*, and *S. typhi* compared to ampicillin, with zones of inhibition of 22 mm, 20 mm, 18 mm, and 17 mm, respectively, except for compounds VIIh and VIIi [142]. Compound VIIa (**114**) showed mild activity with zones of inhibition of 12 mm, 11 mm, 10 mm, and 12 mm, respectively. Compound VIIb (**115**) exhibited inhibition zones of 13 mm, 11 mm, 12 mm, and 11 mm, respectively. Compound VIIc (**116**) showed activity against the four strains, with zones of inhibition of 19 mm, 18 mm, 15 mm, and 15 mm. Compound VIId (**117**) exhibited inhibition zones of 20 mm, 15 mm, 17 mm, and 16 mm against all strains tested. Compound VIIe (**118**) showed activity, with inhibition zones of 23 mm, 21 mm, 20 mm, and 18 mm. Compound VIIf (**119**) exhibited activity, with inhibition zones of 19 mm, 17 mm, 18 mm, and 18 mm, respectively. Compound VIIg (**120**) showed potent activity against all four strains, with zones of inhibition of 24 mm, 22 mm, 21 mm, and 20 mm. Compound VIIh (**121**) showed activity against *B. subtilis* and *S. typhi* only, possibly due to the unsaturation on the phenyl ring. Compound VIIi(**122**) showed activity against *B. subtilis* and *S. aureus* only, with inhibition zones of 13 mm and 10 mm, respectively, possibly due to the three methyl groups substituted on the phenyl ring reducing the activity.

In the same investigation, the antifungal activity of all compounds (VIIa-h) was compared to clotrimazole (10 μg/cup) against *C. albicans* and *A. niger*. Except for VIIb, which showed activity against *A. niger* only, with a zone of inhibition of 12 mm, the compounds exhibited mild to higher antifungal activity. Compounds VIIa, VIIc, VIId, VIIf, VIIh, and VIIi showed moderate activity, with zones of inhibition ranging from 14 to 30 mm against *C. albicans* and from 13 to 20 mm against *A. niger* [143]. In compound **116**, the inclusion of a hydroxyl group at the second position of the phenyl ring improved the inhibitory action, whereas compound **115**’s activity is mild due to its simple phenyl group. Comparing compound **114** to compound **115**, the substitution of a dimethylamino group at the fourth position of the phenyl group enhanced the inhibitory action. The presence of a chloro group at the fourth position of the phenyl group in compound **117** demonstrated about equal activity, whereas compound **118** demonstrated excellent activity by substituting a methoxy group at the fourth position of the phenyl group. Consequently, the activity was raised by the presence of electron-withdrawing groups [143]. Table 2 lists other oxazole derivatives and their antimicrobial activities.

#### 5.1.10. Isoxazole Derivatives

Isoxazole (**123** in Figure 8) and its derivatives are heterocyclic compounds characterized by a five-membered ring containing oxygen and nitrogen atoms at the 1 and 2 positions, respectively. Isoxazole derivatives have significant antimicrobial properties, including antibacterial and antifungal activities [144,145,146,147].

The compounds 2-(Cyclohexylamino)-1-(5-nitrothiophen-2-yl)-2-oxoethyl-5-amino-3-methyl-1,2-oxazole-4-carboxylate (**124** in Figure 8) and 2-(benzylamino)-1-(5-nitrothiophen-2-yl)-2-oxoethyl-5-amino-3-methyl-1,2-oxazole-4-carboxylate (**125** in Figure 8) are two synthesized isoxazole derivatives using a click nucleophilic conjugate addition via an aza-Michael reaction, and the two compounds showed excellent antibacterial activity [131]. Compound **124** showed activity against *S. aureus*, *P. aeruginosa*, and *C. albicans*, with MIC values of 0.00012, 0.125, and 0.063 mg/mL, respectively. Compound **125** showed activity against *S. aureus*, *P. aeruginosa*, and *C. albicans*, with MIC values of 0.00024, 0.063, and 0.02 mg/mL, respectively. It was speculated that the presence of a thiophene ring is the cause of antimicrobial activity in **124** and **125**, and, in addition to the presence of the positive charge of donor S, atoms in the heteroaromatic ring were responsible for damage of the microbial cell wall/membrane, potential leakage of cytoplasmatic content to the environment, and the eventual cellular damage [147].

A series of compounds were synthesized by the cyclocondensation reaction between 1-aryl-3-[5-(4-nitrophenyl)-2-furyl]prop-2-en-1-ones (4a-j) and hydroxylamine hydrochloride in the presence of sodium acetate in glacial acetic acid at the reflux temperature. The compounds with R group substitutions at the third position of isoxazole were investigated for their antimicrobial activities [148]. All compounds (**126**–**131** in Figure 8) showed activity against *Bacillus coccus*, *S. aureus*, *E. aerogenes*, *P. aeruginosa*, and *A. niger* at a concentration of 40 µg/mL, but to varying extents compared to the reference standards amoxicillin, benzyl penicillin, ciprofloxacin, erythromycin, and griseofulvin. Compound **126** was the most active against *A. niger* and *B. coccus*. Compound **130** had similar activity to compound **126** against *B. coccus*, likely due to the simple hydrogen and 4-CH_3_ substituents on the benzene ring at the C3 position of isoxazole. Compounds **128** and **131** were the most active against *S. aureus*, possibly due to the presence of 2-NO_2_ and 4-F substituents.

Regarding *P. aeruginosa*, compounds **127** and **129** were the most active, possibly due to the 4-Cl and 4-OCH_3_ substituents [148]. Table 2 lists other isoxazole derivatives with antimicrobial activity.

#### 5.1.11. Thiazole Derivatives

Thiazole alkaloid (**132** in Figure 8) and its derivatives are a class of heterocyclic compounds characterized by a five-membered ring containing sulfur and nitrogen atoms at the 1 and 3 positions, respectively. Due to their wide range of biological activities, thiazole derivatives are used in medicinal chemistry to develop new therapeutic agents [149,150,151].

A synthesized series of compounds with thiazole rings, coded 7, 13, 17a, 17b, 19c, and 21b (**133**–**138,** respectively, in Figure 8), were investigated for their antimicrobial activities and compared with the efficacy of amphotericin B, ampicillin, and gentamicin at a concentration of 30 µg/mL. The formation of such derivatives proceeded via the reaction of 2-bromo-1-(4-methyl-2-(methylamino)thiazol-5-yl)ethan-1-one with heterocyclic amines, o-aminothiophenol, and thiosemicarbazone derivatives. All compounds showed significant inhibition zones against *S. aureus*, *S. epidermidis*, *B. subtilis*, *E. coli*, *K. pneumoniae*, *S. typhimurium*, *A. niger*, and *Geotrichum candidum* [152].

Compound 17b (**136**) showed high reactivity against the bacterium *E. coli*, with MIC values of 0.49 µg/mL, the same as gentamicin. Additionally, compound 19c (**137**) had a similar MIC (0.98 µg/mL) to amphotericin B against *A. niger*. Compound 21b (**138**) and ampicillin had MIC values of 1.95 µg/mL against *S. epidermidis.* In other words, the antibacterial activity was enhanced by inserting a heterocyclic system at position 5 of the thiazole ring. Additionally, potential activity was greater than that of the reference antibiotics utilized when the system contained several thiazole rings [152].

Sixteen thiazole alkaloid derivatives of N-[4-(substituted)-1,3-thiazol-2-yl]-2-(substituted)acetamide/acetate (**139**–**154** in Figure 9) have been synthesized through a reaction of substituted thiazole carboxylates and piperidines and screened for their in vitro antibacterial and antifungal activity against *S. aureus*, *E. coli*, *P. aeruginosa*, *K. pneumoniae*, *Penicillium marneffei*, *Trichophyton mentagrophytes*, *A. flavus*, and *A. fumigatus* [153].

The results indicate a high level of reactivity (MIC 6.25 µg/mL) against several bacterial strains, including *E. coli* (**139** and **147**), *S. aureus* (**140**, **141**, **143**, **149**, and **151**), *P. aeruginosa* (**140**, **143**, **148**, **151**, and **153**), and *K. pneumoniae* (**144**). Furthermore, the results demonstrate a high level of reactivity (MIC 6.25 µg/mL) against a variety of fungal strains, including compounds **140**, **141**, **145**, **148**, and **151** against *P. marneffei*; compounds **142**, **146**, **148-150**, and **152** against *A. flavus*; compounds **149** and **154** against *A. fumigatus*; and compound **142** against *T. mentagrophytes*. Ampicillin and itraconazole showed MIC values of 6.25 µg/mL against bacterial and fungal strains [153]. Table 2 displays more compounds with antibacterial activity than thiazole derivatives.

#### 5.1.12. Quinazoline Derivative

Quinazoline alkaloids are natural products commonly found in *Nicotiana tabacum*. These compounds are derived from the quinazoline structure, which consists of a benzene ring fused with a pyrimidine ring. Quinazoline (**155** in Figure 9) is a naphthalene molecule where nitrogen atoms replace the carbon atoms at positions 1 and 3 [154,155,156].

Two new quinazoline-4-ones have been developed to address the critical issue of antibiotic resistance. Antimicrobial screening revealed that the compounds 2-(pyrrolidin-1-ylmethyl)-3-[(3,4,5-trimethoxybenzylidene)amino]quinazolin-4(3H)-one and 3-[(4-nitrobenzylidene)amino]-2-(pyrrolidin-1-ylmethyl)quinazolin-4(3H)-one (**156** and **157** in Figure 9) inhibited biofilm formation in *P. aeruginosa*, a quorum-sensing regulated bacterium, at sub-minimum inhibitory concentrations (sub-MICs) with IC50 values of 3.55 and 6.86 µM, respective [157].

Another synthesized quinazoline from starting compounds 2-(4-substituted)Phenyl benzoxazin-4-one and 2-Phenyl-benzoxazin-4-one, were prepared from Anthranilic acid and 4-substituted and unsubstituted benzoyl chloride derivatives in the presence of pyridine. A set of six Schiff bases were synthesized by reacting 2-(4-substituted)Phenyl-3-amino quinazoline-4-3(H)one and 2-Phenyl-3-amino quinazoline -4-3(H)one with various substituted aromatic aldehydes. The synthesized 3-[(E)-(furan-3-ylmethylidene)amino]- 2-phenylquinazolin-4(3H)-one (**158** in Figure 9) was tested for its antimicrobial activity. It was effective against Gram-positive bacteria such as *S. aureus*, with a minimum inhibitory concentration (MIC) of 100 µg/mL and *B. subtilis*, and it featured an MIC of 200 µg/mL. Additionally, it showed activity against Gram-negative bacteria like *E. coli* and *Shigella dysenteriae*, with MIC values of 100 and 200 µg/mL, respectively.

In the same study, 3-(methylidene-amino)-2-phenylquinazolin-4(3H)-one (**159** in Figure 9) demonstrated activity against *S. aureus* and *B. subtilis*, with an MIC of 200 µg/mL. For Gram-negative bacteria, it inhibited *E. coli* and *S. dysenteriae*, with MIC values of 300 and 400 µg/mL, respectively.

Furthermore, 2-(4-fluorophenyl)-3-[(E)-(2-hydroxybenzylidene)amino]quinazolin- 4(3H)-one (**160** in Figure 9) was synthesized and evaluated for its antibacterial and antifungal activities. It exhibited antibacterial activity against *S. aureus*, with an MIC of 12.5 µg/mL, and antifungal activity against *C. albicans*, *Cryptococcus neoformans*, *Sporothrix schenckii*, *T. mentagrophytes*, and *A. fumigatus*, with MIC values of 6.25, 12.5, 50, 50, and 25 µg/mL, respectively [158,159]. Table 2 shows other quinazoline derivatives.

#### 5.1.13. Quinoline Derivatives

Quinoline (**161** in Figure 9) is an alkaloid characterized by a benzene ring fused to a pyridine ring, similar to isoquinoline, but differing in terms of the nitrogen atom’s position. The compound 4-hydroxymethyl-quinoline (**162** in Figure 9) was extracted from *myxobacterium Labilithrix luteola* and tested for its antimicrobial activity. It was found to be active against *C. albicans*, with a minimum inhibitory concentration (MIC) of 33.3 µg/mL [160,161,162].

In terms of two synthesized compounds, 2-{2-[(2-chloroquinolin-3-yl)methylene]hydrazinyl} acetonitrile and 6-(2-chloroquinolin-3-yl)-4-(4-aminophenyl)pyrimidin-2(1H)-one (**163** and **164** in Figure 9), **163** was synthesized by ethanol refluxed of 2-chloroquinoline-3-carbaldehyde with glacial acetic acid and 2-hydrazinyl acetonitrile, while **164** was synthesized by ethanol refluxed of chalcone with urea and hydrochloric acid. The two compounds demonstrated significant potency against various bacterial strains, including *S. aureus*, *B. subtilis*, and *E. coli*, when compared to ciprofloxacin, ampicillin, and gentamicin, with MIC values ranging from 0.24 to 0.98 µg/mL. Additionally, these compounds exhibited strong antifungal activity against *A. fumigatus*, *Scedosporium apiospermum*, and *C. albicans*, with MIC values ranging from 0.24 to 0.98 µg/mL, comparable to amphotericin B [163].

The compound 4-methyl-2-(4-chlorophenyl)quinolone (**165** in Figure 9) is a synthesized from aniline and benzaldehyde derivatives, with acetone in hydrochloric acid acting as a medium. The compound was tested for its antibacterial activity. It showed significant activity against *E. coli* and *P. aeruginosa* compared to the standard drug Streptomycin, with zones of inhibition measuring 28 and 35 mm, respectively. However, it did not exhibit strong activity against *S. aureus* [164]. More quinoline derivatives are presented in Table 2.

#### 5.1.14. Acridine Derivatives

Acridine (**166** in Figure 10) is an anthracene polycyclic compound where a nitrogen group replaces a central CH group. Kuanoniamine D (**167** in Figure 10) is a natural compound found in *Cystodytes*, *Cystodytes dellechiajei*, and *Oceanapia*. When extracted from *Cystodytes dellechiajei*, it exhibited antibacterial activity against *E. coli* and *M. luteus*, with minimum inhibitory concentration (MIC) values ranging from 2.2 to 17.4 µM [165,166].

The acridine derivative 1-benzoyl-4-(7-benz[c]acridinyl) thiosemicarbazide (**168** in Figure 10) was designed and studied for its antimicrobial activity. This compound showed excellent antibacterial and antifungal activity, inhibiting the growth of *S*, *aureus*, *Staphylococcus castellani*, *E. coli*, *P. aeruginosa*, and *C. albicans*, with MIC values of 10, 10, 20, 10, and 10 µM, respectively [167,168].

The antimicrobial activity of the synthesized novel acridine derivatives, 10-[3′-(N,N-diethylamino)propyl]acridin-9-one and 10-[3′-(N-pyrrolidin-2″-one)propyl] acridin-9-one (**169** and **170** in Figure 10) were investigated. The N10-Alkylated acridin-9-ones were synthesized in two series; in the first series, N10-alkylation with 1-bromo-3-chloro propane and, in the second series, 1, 2-dichloroethane were achieved by using tetrabutyl ammonium bromide as a phase transfer catalyst. These compounds showed remarkable activity against *B. subtilis*, *S. aureus*, *E. coli*, *P. aeruginosa*, *Shigella sp.*, *C. albicans*, and *A. niger*, with zones of inhibition ranging from 14 to 28 mm. Compound **169** exhibited more potent antibacterial activity against the tested pathogens, while compound **170** demonstrated stronger antifungal activity against *C. albicans* than the standard drug clotrimazole [168,169,170]. Additional acridine derivatives are presented in Table 3.

#### 5.1.15. Indole Derivatives

Tryptophan is the precursor of indole alkaloids, which are bicyclic compounds featuring a nitrogen-containing pyrrole ring fused to a six-membered benzene ring. The indole structure (**199** in Figure 11) includes a five-membered ring with a nitrogen atom [180,181]. Naucleidinal, extracted from *Nauclea latifolia*, was studied for its antimicrobial activity against *Haemophilus influenzae*. Among the compounds extracted from *N. latifolia*, naucleidinal exhibited the most potent activity with a minimum inhibitory concentration (MIC) of 3.1 µg/mL [172]. Caulilexin A (**200** in Figure 11) is an indole extracted from cauliflower (*Brassica oleracea var. botrytis*). It was isolated to evaluate its antimicrobial activity and was found to inhibit the growth of *Leptosphaeria maculans*, *S. sclerotiorum*, and *Rhizoctonia solani* by 55%, 100%, and 100%, respectively [182].

The compound 2,5,6-Tribromo-3-[(3′-bromo-4′-hydroxyphenyl)methyl]-1H-indole (**201** in Figure 11) and 5,6-dibromo-1-hydroxy-3-isopropenyl-indole-2-one (**202** in Figure 11) are natural indole derivatives extracted from *Laurencia similis*. These compounds were tested for their antimicrobial activity and found to be effective against several bacteria. Compound **201** showed the most potent activity against seven tested bacteria, inhibiting the growth of *B. subtilis*, *Bacillus thuringiensis*, *S*, *aureus*, *Agrobacterium tumefaciens*, *Pseudomonas lachrymans*, *Ralstonia solanacearum*, and *Xanthomonas vesicatoria*, with MIC values ranging from 2 to 8 µg/mL. Compound **202** exhibited antibacterial activity against *S. aureus* and *A. tumefaciens* with an MIC of 12.5 µg/mL [183]. While 6,7-dibromo-1,3-dimethyl-9H-carbazole and 5-dibromo-2-methoxycarbonylamino-benzoate ester did not show any antibacterial activity when evaluated in the same study. Thus, we may conclude that the number and the position of the halogen and the presence of the hydroxy group had substantially different influences on the growth of microorganisms. Additional indole derivatives are listed in Table 3.

#### 5.1.16. Imidazole Derivatives

Imidazole (**203** in Figure 11) is a compound featuring a five-membered ring with two nitrogen atoms at positions 1 and 3. Imidazole alkaloids exhibit various biological activities, including antimicrobial and antifungal properties [184,185,186]. A synthesized imidazole compound (**204** in Figure 11) derived from L-valine and L-phenylalanine showed potent antibacterial activity against *Escherichia coli* and *B. subtilis* with minimum inhibitory concentration (MIC) values of 32 and 4 µg/mL, respectively [187].

Another study that synthesized imidazole derivatives found that compounds **205-207** (Figure 11) have notable antimicrobial activity [175]. Compound **205** ((2S)-2-Amino-3-(1-((E)-1-((E)-2-benzylidenehydrazinyl)-3,7-dimethylocta-2,6-dien-1-yl)-1H-imidazol-4-yl)propanoic acid) showed remarkable antibacterial activity against *K. pneumoniae* with an MIC value of 0.25 µg/mL, compared to the standard drug ciprofloxacin, which had an MIC value of 32 µg/mL because the citral connected with imidazole moiety with 2-amino acetic acid. Compound **206** ((2S)-2-Amino-3-(1-(((E)-2-benzylidenehydrazinyl)(pyridin-4-yl)methyl)-1H-imidazol-4-yl)propanoic acid) was more active against *S*, *aureus*, with an MIC value of 0.25 µg/mL, compared to ciprofloxacin, which had an MIC value of 0.5 µg/mL, due to the pyridine ring moiety with imidazole moiety and 2-amino acetic acid. Compound **207** ((2S)-2-Amino-3-(1-(((E)-2-benzylidenehydrazinyl)(4-chlorophenyl)methyl)-1H-imidazol-4-yl)propanoic acid) exhibited greater antifungal activity against *C. albicans* with an MIC value of 0.25 µg/mL, compared to clotrimazole, which had an MIC value of 0.5 µg/mL because 4-chloro phenyl and imidazole moiety with 2-amino acetic acid [175]. Table 3 lists additional imidazole derivative compounds.

#### 5.1.17. Purine Derivatives

Purine (**208** in Figure 11) is a bicyclic compound with an imidazole ring fused to a pyrimidine ring. Pyrimidine is a six-membered ring containing two nitrogen atoms. Purine can be found in *Homo sapiens*, *Panax ginseng*, and *Bos taurus* [188,189,190]. Two novel purine derivatives were synthesized to study their antimicrobial activity: 1,3-dimethyl-7-{2-[(4-methylphenyl)amino]ethyl}-8-nitro-3,7-dihydro-1H-purine-2,6-dione and 1,3-dimethyl-8-nitro-7-[2-(phenylamino)ethyl]-3,7-dihydro-1H-purine-2,6-dione (**209** and **210** in Figure 11). These compounds exhibited good antibacterial activity compared to chloramphenicol against *E. coli*, *P. aeruginosa*, *B. subtilis*, and *S. aureus*, with zones of inhibition ranging from 15.87 to 22.65 mm. They also showed good antifungal activity compared to nystatin against *Aspergillus niger* and *C. albicans*, with zones of inhibition ranging from 22.78 to 30.38 mm [191]. Caffeine (**211** in Figure 11) is a purine derivative that exhibited antimicrobial activity against *E. coli* at concentrations ≥ 0.50%. When extracted from *Coffea robusta*, it showed antibacterial activity against both Gram-positive and Gram-negative bacteria, including *S. typhi*, *P. aeruginosa*, *S. aureus*, *B. cereus*, and *Lactobacillus bulgaricus* [192]. Additional purine derivatives are included in Table 3.

### 5.2. Alkaloids with Nitrogen in the Side Chain (Protoalkaloids)

Protoalkaloids contain a nitrogen atom outside the ring, retained as a side chain rather than as part of the heterocyclic system. These compounds can be produced from biogenic amines or amino acids. Examples of protoalkaloids include β-Phenylethylamine, muscarine, benzylamine, and colchicine. Protoalkaloids can be derived from L-tyrosine and L-tryptophan [193,194].

#### 5.2.1. β-Phenylethylamine Derivatives

β-Phenylethylamine derivatives are compounds that act as monoamine alkaloids, characterized by their phenethylamine backbone, which consist of a phenyl group (a benzene ring) attached to an ethylamine chain [195]. These derivatives can undergo modifications such as changes in the alkyl side chain length, substitutions on the aromatic ring, or alterations to the amine portion of the molecule. Such modifications can enhance the compound’s biological activity, pharmacokinetics, or selectivity for specific receptors. β-Phenylethylamine derivatives have been studied for various applications, including antimicrobial activities against pathogens [196].

A study investigated the antimicrobial effects of phenylethylamine (compound **212** in Figure 11) against the growth of *Listeria monocytogenes* in various media. Phenylethylamine was found to inhibit the growth of *L. monocytogenes* in a concentration- and strain-dependent manner, with significant growth inhibition being observed at concentrations as low as 1.56 mg/mL. The minimum inhibitory concentrations (MICs) across 62 strains ranged from 8 to 12.5 mg/mL. Phenylethylamine not only prolonged the lag phase and reduced the growth rate but also exhibited bactericidal effects, reducing bacterial counts by 1 to 8 log CFU/mL in a strain-specific manner. The strain *L. monocytogenes* EGDe was the most sensitive, with the lowest MIC. Phenylethylamine’s inhibitory effects remained stable under simulated heat treatment conditions, indicating its potential as a robust antimicrobial agent. The study also found that phenylethylamine inhibited bacterial respiration or metabolic activity at concentrations equivalent to or above the MIC [197].

Synephrine **(213** in Figure 11) is a phenethylamine derivative with two hydroxyl groups that increase its polarity and biological activity. It is characterized by a benzene ring (phenyl group) connected to an ethylamine chain. One hydroxyl group is linked to the carbon next to the amine group, forming a chiral center, while the other is placed on the benzene ring at the meta position. Due to its structural similarity to the neurotransmitter adrenaline, synephrine plays a vital role in several biological processes [198]. Extracted from *Citrus aurantium*, synephrine exhibits significant antifungal activity against various molds and yeast strains such as *A. flavus*, *Arthroderma insingulare*, *Alternaria alternate*, and *Penicillium* spp., with colony diameters that are significantly reduced compared to the control. Specifically, when treated with synephrine, the colony diameter of *A. flavus* reduced from 9.76 mm to 6.76 mm. Additionally, while synephrine slightly increased the colony size of *Candida* spp., it was still identified as having antifungal potential [199].

Hordenine (**214** in Figure 11) is a notable phenethylamine derivative with an ethylamine chain linked to a benzene ring and *N*-methyl group bonded to the side chain’s nitrogen atom. This structural characteristic contributes to its distinct biological action and differentiates it from its parent compound, phenethylamine [200]. Hordenine’s solubility and interaction with biological receptors are influenced by the absence of a hydroxyl group on the phenyl ring, in contrast to similar substances [201]. At a concentration of 1000 µg/mL, hordenine demonstrated antibacterial activity against *S. aureus* and MRSA, indicating that, while hordenine may not inhibit bacterial growth in a disk diffusion setup, it shows some effectiveness at higher concentrations in a more direct interaction assay [202].

#### 5.2.2. Colchicine Alkaloids

Colchicine derivatives are tricyclic alkaloids characterized by a tropolone ring (a seven-membered aromatic ring containing one oxygen atom), an aromatic benzene ring, and an acetamide group. The molecule also features three methoxy groups attached at specific ring positions. The nitrogen atom in the acetamide group is connected to the C7 position of the tropolone ring [203].

Colchicine (**215** in Figure 11), derived from *Gloriosa superba* Linn, has demonstrated notable antimicrobial activity against various fungal and bacterial pathogens. The crude methanol extract and its fractions were evaluated for their effects on several fungi, including *Trichophyton longifusus*, *C. albicans*, *A. flavus*, and *Microsporum canis*. The extract showed significant inhibition of *T. longifusus* and *M. canis*, with the n-butanol fraction displaying the highest activity. Against *C. albicans* and *C. glabrata*, the n-butanol and ethyl acetate fractions were the most effective, showing up to 90% inhibition. The extract exhibited intense antibacterial activity against *S. aureus*, particularly in the n-butanol fraction. However, the plant showed low to no activity against other bacterial strains, such as *E. coli* and *P. aeruginosa*. The study suggests that *Gloriosa superba* Linn, and, consequently, colchicine, could be a potential natural antifungal and antibacterial agent, though further research is needed to understand the exact mechanisms and broader applications [204].

Demecolcine (**216** in Figure 11), also known as colcemid, is a derivative of colchicine where a hydrogen atom replaces the methoxy group on the tropolone ring (at position C7 [205]. This slight modification alters its binding properties to tubulin compared to colchicine [206]. Demecolcine, derived from *Colchicum balansae*, has been studied for its antibacterial properties. The ethanol extract of the plant demonstrated a weak inhibitory effect against a variety of bacteria, including *S. aureus*, *S. epidermidis*, *E. faecalis*, *K. pneumoniae*, *E. coli*, *E. cloacae*, *Serratia marcescens*, *P. vulgaris*, *P. aeruginosa*, and *Salmonella typhimurium*. Among these, *S. aureus* ATCC 25923 was the most sensitive, showing a 10 mm inhibition zone. However, compared to standard antibiotics, the antimicrobial activity of the ethanol extract was relatively lower, indicating the limited efficacy of demecolcine as an antibacterial agent [207].

Colchicoside (**217** in Figure 11) is a glycosylated colchicine derivative featuring a sugar moiety (glucose) attached to the colchicine backbone. The core tricyclic structure of colchicine is preserved with the addition of a glycoside linkage [208].

A study conducted on the *Colchicum* species revealed that colchicoside, a tropolone alkaloid, is one of the main compounds found in the corms of various *Colchicum* species, including *C. speciosum*, *C. autumnale*, and *C. robustum*. Colchicoside was present in significant quantities, particularly in *C. speciosum* (46.35 mg/100 g corm). The study emphasized that these *Colchicum* species possess bioactive compounds, like colchicoside, with potential therapeutic effects, including anti-inflammatory properties, making them valuable for treating conditions such as arthritis. However, the study did not explicitly detail the antimicrobial activity of colchicoside [209].

#### 5.2.3. Benzylamine Derivatives

Benzylamine alkaloids are a group of organic compounds that contain a benzylamine moiety, which consists of a benzyl group attached to an amine functional group. These compounds can be modified to create derivatives with different biological properties [210,211].

Capsaicinoids, primarily capsaicin and dihydrocapsaicin (**218** and **219** in Figure 11), are the compounds responsible for the pungency of chili peppers, comprising about 90% of the capsaicinoid content. The capsaicin molecule exists predominantly in its trans-isomer form due to steric hindrance in the cis form. Capsaicinoids are synthesized in chili fruit placentas via enzymatic condensation of vanillylamine (**220** in Figure 11) with fatty acids of varying chain lengths [212,213]. These compounds have significant applications in the food and pharmaceutical industries due to their antioxidant, anticarcinogenic, and anti-inflammatory properties and their role in promoting energy metabolism. The antimicrobial activity of capsaicin was evaluated using well diffusion and dilution assays. In the well diffusion assay, a 2 cm zone of inhibition around the capsaicin sample indicated significant antimicrobial activity against *B. subtilis*, demonstrating capsaicin’s ability to inhibit bacterial growth. The dilution test further supported this, showing a 40% decline in *B. subtilis* growth after 48 h of incubation with capsaicin. Although no immediate effects were observed, the inhibition became apparent after 36 h, confirming capsaicin’s potential as an antimicrobial agent [214].

Dihydrocapsaicin, isolated from *Capsicum frutescens*, demonstrated significant antimicrobial activity against various microorganisms, including Gram-positive and Gram-negative bacteria and *C. albicans*. The MIC values for dihydrocapsaicin ranged from 0.6 to 5 µg/mL for Gram-positive bacteria and 2.5 to 5 µg/mL for Gram-negative bacteria, indicating more substantial effectiveness against Gram-positive bacteria, likely due to their simpler cell wall structure. Notably, dihydrocapsaicin also inhibited the growth of *C. albicans* at a lower MIC of 10 µg/mL, highlighting its broad-spectrum antimicrobial potential [215,216].

Vanillylamine has promising antimicrobial properties and is particularly interesting due to its potential to inhibit many bacterial and fungal strains. It showed significant activity against *S. aureus* and *B. cereus*, with MIC values ranging from 8 to 32 µg/mL, and against *E. coli* and *P. aeruginosa*, with MIC values ranging from 16 to 64 µg/mL. In addition, it has antifungal activity against *C. albicans*, with MIC values ranging from 16 to 64 µg/mL [217,218].

### 5.3. Polyamines Alkaloids

Polyamine-derived alkaloids are a fascinating group of nitrogen-containing compounds derived from polyamines like putrescine, spermidine, and spermine. These compounds exhibit antimicrobial properties, making them relevant in developing new antimicrobial agents [219,220].

A synthesized putrescine bisamide compound, N,N’-(Butane-1,4-diyl) bis(3-(benzo[d][1,3]dioxo]-5-yl) propanamide (**221** in Figure 11), showed significant antibacterial activity against *S. typhi*, *V. cholerae*, *S. dysenteriae*, *E. faecalis*, and *S. aureus*. The inhibitory zones ranged between 10 and 14 mm, with an MIC value of 1 μg/mL against *S. typhi* and 10 μg/mL against the others. The MIC of 1 μg/mL was comparable to the potency of ciprofloxacin, used as a positive control [221]. Spermidine-capped carbon dots (S-PCDs) exhibit notable antimicrobial activity against Gram-positive and Gram-negative bacteria. They show potent efficacy against *S. aureus*, with an MIC value of 16 µg/mLn and are effective against multidrug-resistant strains such as *E. coli*, *Shewanella putrefaciens*, *P. aeruginosa*, and *Vibrio parahaemolyticus*, with MIC values of 64, 32, 64, and 512 μg/mL, respectively [222,223].

The antifungal properties of dicaffeoyl spermidine and N1, N5, N10-tri-p-(E, E, E)-coumaroylspermidine (**222** and **223** in Figure 12) inhibited the development of the oat leaf stripe pathogen *Pyrenophora avenae*’s mycelial growth and the infection of barley by the powdery mildew fungus *Blumeria graminis* [224]. Additionally, ptilomycalin A (**224** in Figure 12) suppressed *C. neoformans*’ melanogenesis in vitro with an IC50 of 7.3 µM by blocking laccase production in the melanin biosynthetic pathway. It also showed antifungal activity against *C. albicans*, with an MIC of 0.8 µg/mL [225].

In a dose-dependent manner, the spermine derivative compound Budmunchiamine A (**225** in Figure 12) significantly suppressed the growth and fumonisin B1 synthesis by *F. verticillioides*, with an MIC value of 0.125 mg/mL and minimum fungicidal concentrations of 0.25 mg/mL. Furthermore, with MIC values of 2, 1, and 1 µg/mL, respectively, crambescidine 800 (**226** in Figure 12) significantly suppressed the growth of *A. baumannii*, *K. pneumoniae*, and *P. aeruginosa* bacteria [226].

### 5.4. Pseudoalkaloids

Pseudoalkaloids are heterocyclic alkaloids that contain nitrogen but are not produced from amino acids. They are often derivatives of acetate, pyruvic acid, adenine/guanine, or geraniol. They exhibit significant structural variability and potential bioactive properties, particularly regarding their antimicrobial activity [227,228,229].

#### 5.4.1. Steroidal Alkaloids

The two compounds, Epipachysamine-E-5-ene-4-one and iso-N-formyl chonemorphine (**227** and **228** in Figure 12), isolated from *Sarcococca brevifolia*, exhibited robust antibacterial activity against *B. cereus*, *K. pneumoniae*, *S. aureus*, and *P. aeruginosa*, with MICs ranging from 0.0312 to 0.2500 mg/mL. The standard antibiotics amoxicillin and ampicillin had MIC values of 0.0625 and 0.2500 mg/mL, respectively [230,231].

Saligcinnamide (**227** in Figure 12) possessed potent antibacterial activity against *K. pneumoniae*, *P. aeruginosa*, and *S. aureus*, with zone inhibitions of 10.5, 7.5, and 11 mm, respectively, compared with amoxicillin, which had zone inhibitions ranging from 15 to 17 mm. Epipachysamine D (**228** in Figure 12) showed antibacterial activity against *P. mirabilis*, *S. aureus*, and *S. pyogenes*, with zone inhibitions of 8, 12, and 6 mm, respectively, compared with ampicillin, which had zone inhibitions ranging from 12 to 17 mm [232].

Hookerianamide I and hookerianamide H (**229** and **230** in Figure 12) are two compounds extracted from *Sarcococca hookeriana* that inhibited the growth of *B. subtilis*, with MIC values of 69.4 and 323.8 µM, while they inhibited the growth of *M. luteus*, with MIC values of 34.7 and 323.8 µM, respectively. Sarcovagine C, vagenine A, and chonemorphine (**231**–**233** in Figure 12) showed significant antibacterial activity against *B. subtilis*, with MIC values of 64.3, 64, and 90.3 µM, compared with the study positive control standard ampicillin, with an MIC value of 89.7 µM [233].

With MIC values of 120 and 80 μg/mL for triadimefon as a positive control against the phytopathogenic fungi *Phytophthora capsici* and *Rhizoctonia cerealis*, the antifungal activity of the isolated compounds from *Veratrum taliense*, Neoveratalines A and B (**234** and **235** in Figure 12) was studied. The two compounds showed moderate antifungal inhibitory activity against the two phytopathogens with 200 μg/mL MIC values. Verazine alkaloids, veramitaline, theophylline B, theophylline B-3-O-β-D-glucopyranoside, and veramiline-3-O-β-D-glucopyranoside (**236**–**239** in Figure 12) showed antifungal activities against *P. capsici*, with MIC values of 120, 160, 80, and 160 μg/mL, respectively. Additionally, except for veramiline-3-O-β-D-glucopyranoside, they showed antifungal activities against *R. cerealis*, with MIC values ranging from 120 to 160 μg/mL. Moreover, Jerveratrum alkaloids, jervine and jervine-3-O-β-D-glucopyranoside (**240** and **241** in Figure 12) exhibited stronger antifungal activities against *P. capsici*, with MICs at 80 and 120 µg/mL [234,235].

#### 5.4.2. Diterpenes Alkaloids

Several diterpenes demonstrate antimicrobial activity. Sinchiangensine and its analog, lipodeoxyaconitine (**242** and **243** in Figure 12) exhibited antibacterial activity against *S. aureus*, with MIC values of 0.147 and 0.144 mmol/mL, respectively [236,237]. Carmichaedine (**244** in Figure 12), extracted from *Aconitum carmichaelii* root, showed antimicrobial activity against *B. subtilis*, with an MIC of 8 mmol/mL [238].

Moreover, with MIC values of 200, 400, and 800 mg/mL, respectively, the isolated aconicaramide (**245** in Figure 12) from *A. carmichaelii* demonstrated antibacterial activity against *Macrococcus caseolyticus*, *S. epidermidis*, and *S. aureus* [239].

Conversely, Oleracein E (**246** in Figure 12) showed significant antibacterial activity against *S. aureus*, *M. caseolyticus*, *K. pneumoniae*, and *S. pneumoniae*, with MIC values ranging from 50 to 200 mg/mL. In addition, the aconitine derivatives, compounds **247** and **248** in Figure 12, which were obtained from the roots of *Aconitum duclouxii*, showed antifungal activity against *C. albicans*, with MIC values of 51.84 and 128 mg/mL, respectively, while compound **247** also showed antibacterial activity against *B. subtilis*, with an MIC of 147.73 mmol/mL [240,241].

As a conclusion to this chapter, we can state that, based on the research, isoquinoline alkaloids and indole alkaloids stand out as being very promising for use in medicine because of their potent activities against harmful microorganisms.

The most antibacrial and antifungal alkaloid activities are summarized in Table 4.

## 6. Future Implications for Healthcare

Antimicrobial resistance (AMR) has broad and complex future healthcare implications that will make managing infectious diseases extremely difficult. Decades of medical advancement could be jeopardized by the emergence of multidrug-resistant (MDR) bacterial and fungal strains, which could result in a return of illnesses that were previously thought to be treatable. The widespread and frequently improper use of antibiotics in veterinary, agricultural, and healthcare settings aggravates this issue by hastening the evolution of resistance strains [242,243,244].

The healthcare system may have to deal with the possibility of regular medical treatments becoming riskier as AMR continues to rise because of the increased incidence of infections that are challenging to treat. These resistant bacteria are especially dangerous to several populations, including cancer patients, immunocompromised people, and surgical patients [243]. The financial ramifications are also concerning; estimates indicate that, by 2050, the worldwide expense of antimicrobial resistance (AMR) may amount to trillions of dollars, severely taxing healthcare systems, particularly in low- and middle-income nations [242].

Ongoing efforts to combat AMR are essential in response to these concerns. Stewardship programs are one possible approach. Healthcare organizations are increasingly putting antimicrobial stewardship plans into place with the goal of optimizing the use of antibiotics. In order to reduce needless exposure, these programs aim to guarantee that patients receive the proper antibiotic at the right dose for the right amount of time [245,246].

On the other hand, the discovery of novel antimicrobial agents and alternative treatments is the main focus of genomics and the pharmaceutical sector. Promising research topics include the investigation of antimicrobial peptides and innovations like bacteriophage therapy, which employs viruses to target particular bacteria. The development of vaccinations against common bacterial pathogens is a critical strategy in reducing antimicrobial resistance (AMR), and vaccines can also significantly reduce the occurrence of illnesses, hence minimizing the demand for antibiotics [247,248]. Additionally, alkaloids have the ability to target and disrupt vital bacterial processes such protein synthesis, DNA replication, and cell membrane integrity, which lays the groundwork for the creation of novel treatment approaches [14,20].

Additionally, alkaloids’ ability to improve the effectiveness of currently available antibiotics through processes such efflux pump inhibition suggests that they are useful adjuncts in antibiotic therapies [22,27]. Alkaloids’ development as antimicrobial agents will be aided by further research and comprehension of their structure–function interactions. Their varied approaches to combat distinct diseases increase their applicability and provide a way around the present problems caused by antibiotic resistance in diverse healthcare environments [34].

Furthermore, as data sharing globally can help track resistance patterns and improve treatment guidelines, global surveillance, reporting, and upgraded surveillance systems are required to monitor the establishment and spread of resistant strains. Governments can also play a crucial role in controlling the overuse of these essential drugs by enforcing stronger laws to govern the use of antibiotics in healthcare and agriculture [249,250].

## 7. Conclusions

Throughout the study and analysis of alkaloids’ antimicrobial potential, we have highlighted their numerous pharmacological activities and their encouraging potential for pathogen control, especially in light of the growing resistance to traditional antibiotics. We demonstrate the breadth and depth of the important alkaloid classes—true alkaloids, protoalkaloids, polyamine alkaloids, and pseudoalkaloids—and their distinct modes of action by reviewing them. This is especially important when it comes to combating drug-resistant strains. The findings highlight the significance of these organic compounds in improving healthcare and call for further investigation into their processes and applications.

Beyond the current pharmacological applications, the implications of this research point to a paradigm shift in our approach to infectious illness management and medication development. With their innovative methods for increasing the effectiveness of currently available antimicrobial medications and developing new treatments, alkaloids provide a glimmer of hope in the face of the issues brought on by antibiotic resistance.

As the discipline of pharmacology continues to expand, the study of alkaloids in pathogen treatment is a testament to the ability of natural chemicals to impact medical practice. It is an essential first step in tackling some of the most significant problems facing world health today.

## Figures and Tables

**Figure 1 toxins-16-00489-f001:**
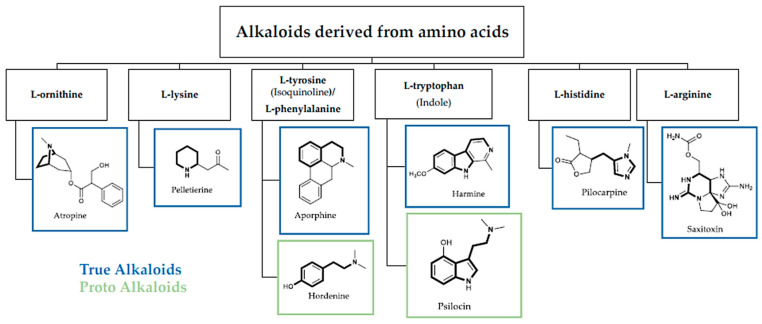
Amino acid-derived alkaloids, as well as true and proto alkaloids. Reproduced with permission from Casciaro, B.; Mangiardi, L.; Cappiello, F.; Romeo, I.; Loffredo, M. R.; Iazzetti, A.; Calcaterra, A.; Goggiamani, A.; Ghirga, F.; Mangoni, M. L.; Botta, B.; Quaglio, D., *Molecules*; published by MDPI, 2020 [20].

**Figure 2 toxins-16-00489-f002:**
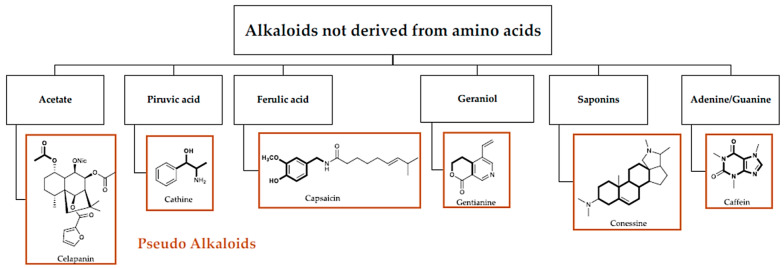
Non-amino acid-derived alkaloids, as well as true and proto alkaloids. Reproduced with permission from Casciaro, B.; Mangiardi, L.; Cappiello, F.; Romeo, I.; Loffredo, M. R.; Iazzetti, A.; Calcaterra, A.; Goggiamani, A.; Ghirga, F.; Mangoni, M. L.; Botta, B.; Quaglio, D., *Molecules*; published by MDPI, 2020 [20].

**Figure 3 toxins-16-00489-f003:**
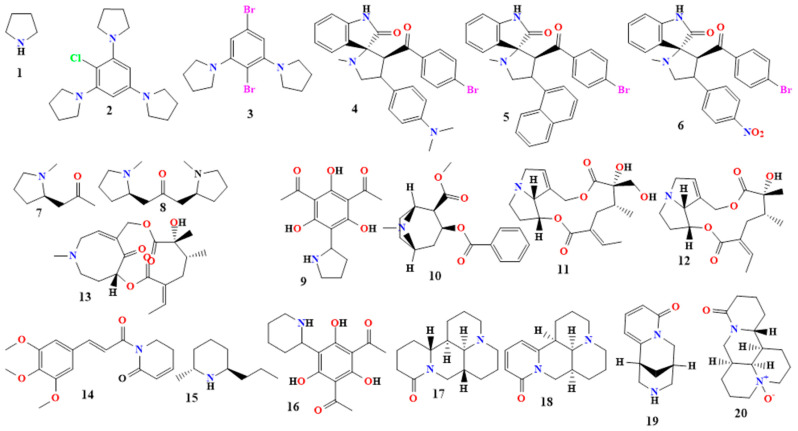
Chemical structures of alkaloids **1**–**20**.

**Figure 4 toxins-16-00489-f004:**
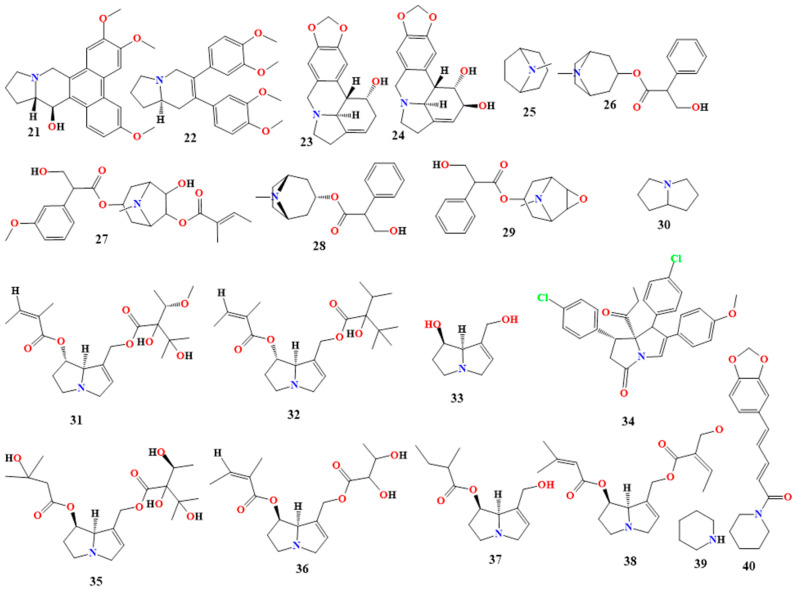
Chemical structures of alkaloids **21**–**40**.

**Figure 5 toxins-16-00489-f005:**
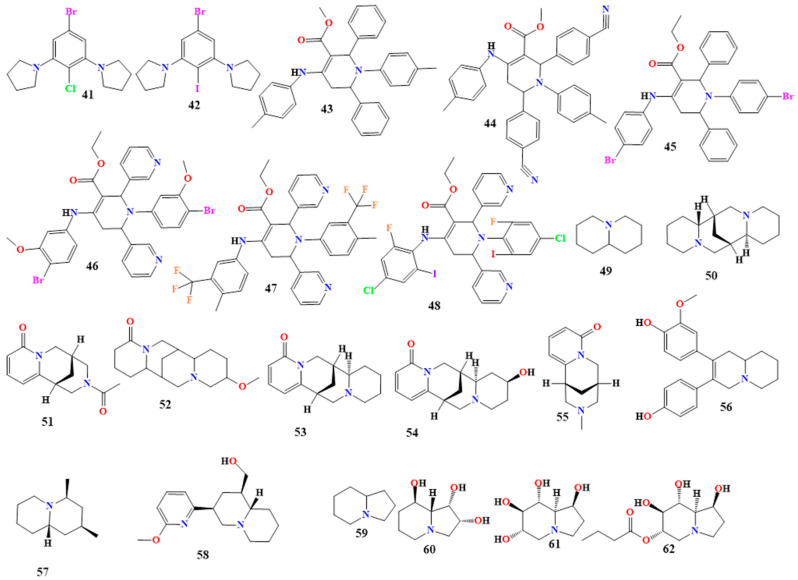
Chemical structures of alkaloids **41**–**62**.

**Figure 6 toxins-16-00489-f006:**
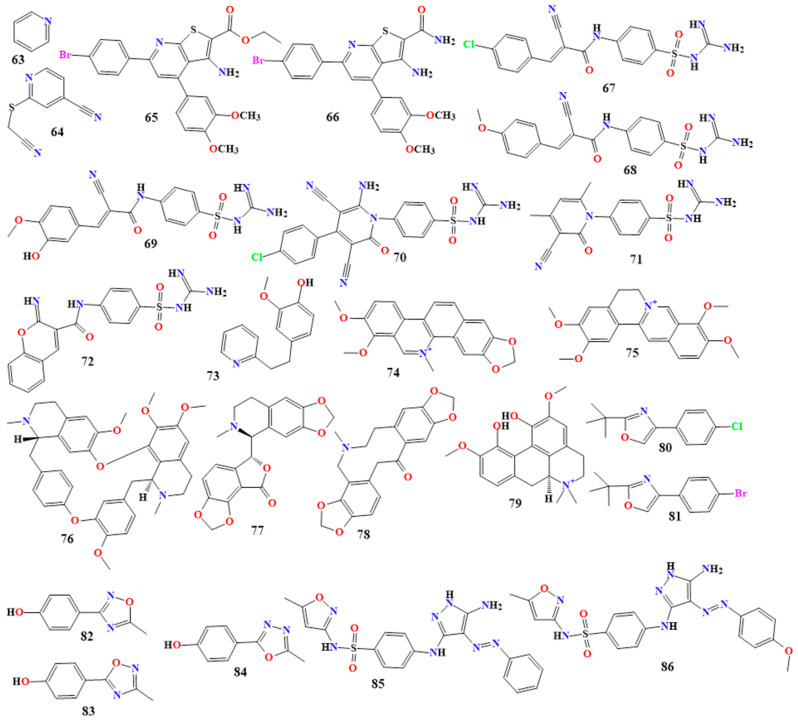
Chemical structures of alkaloids **63**–**86**.

**Figure 7 toxins-16-00489-f007:**
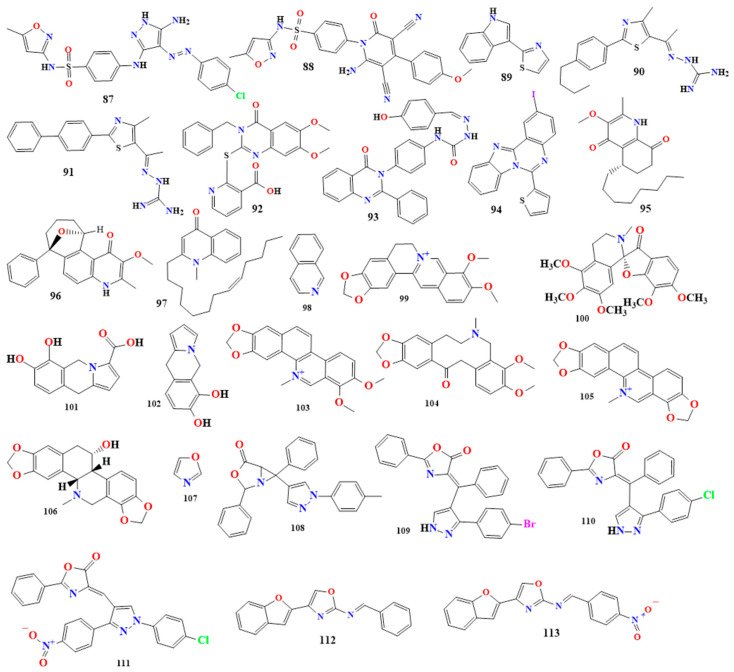
Chemical structures of alkaloids **87**–**113**.

**Figure 8 toxins-16-00489-f008:**
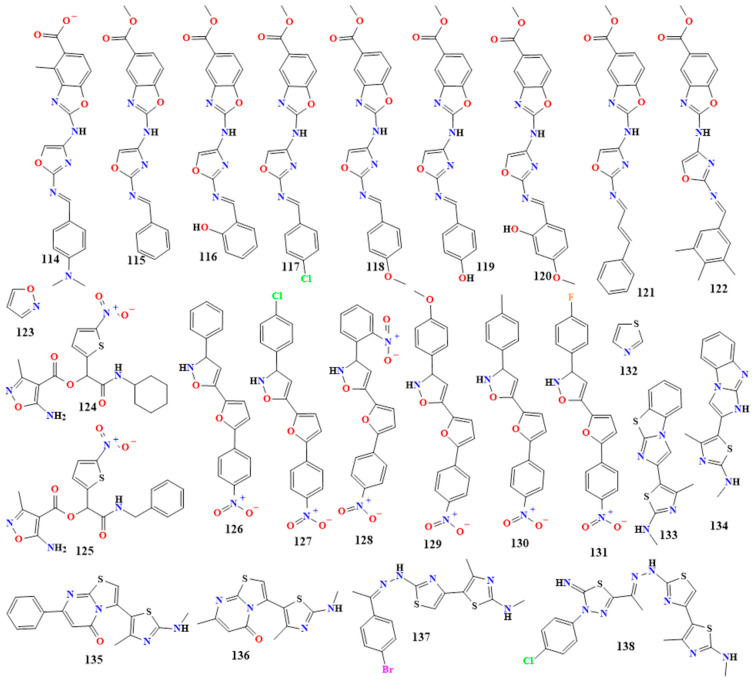
Chemical structures of alkaloids **114**–**138**.

**Figure 9 toxins-16-00489-f009:**
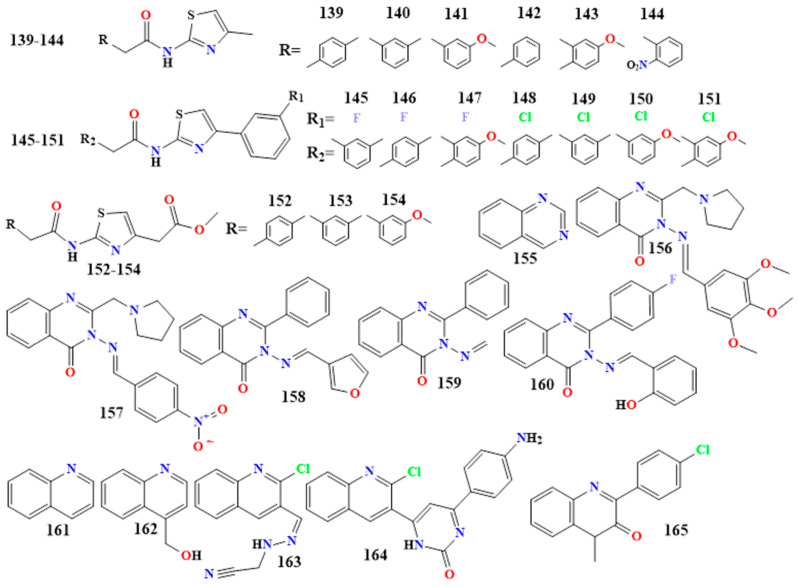
Chemical structures of alkaloids **139**–**165**.

**Figure 10 toxins-16-00489-f010:**
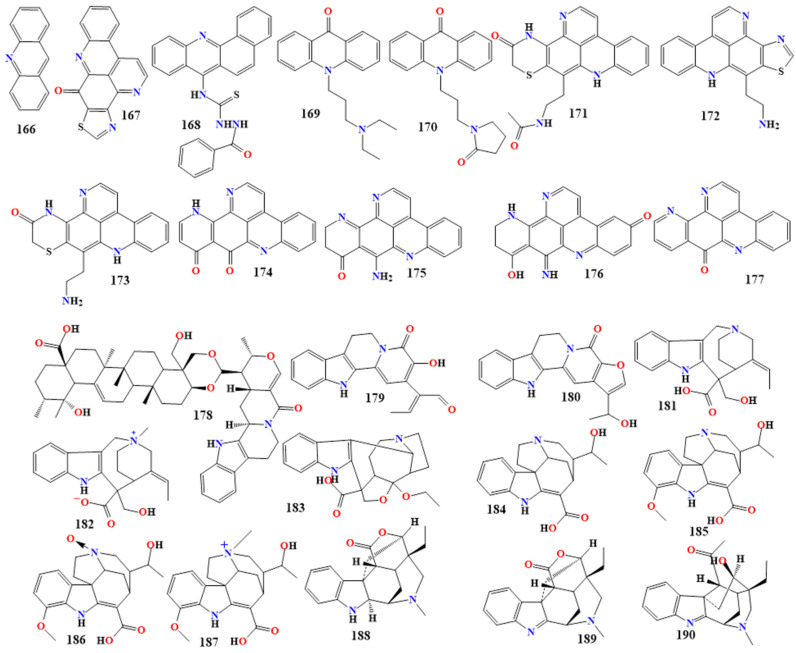
Chemical structures of alkaloids **166**–**190**.

**Figure 11 toxins-16-00489-f011:**
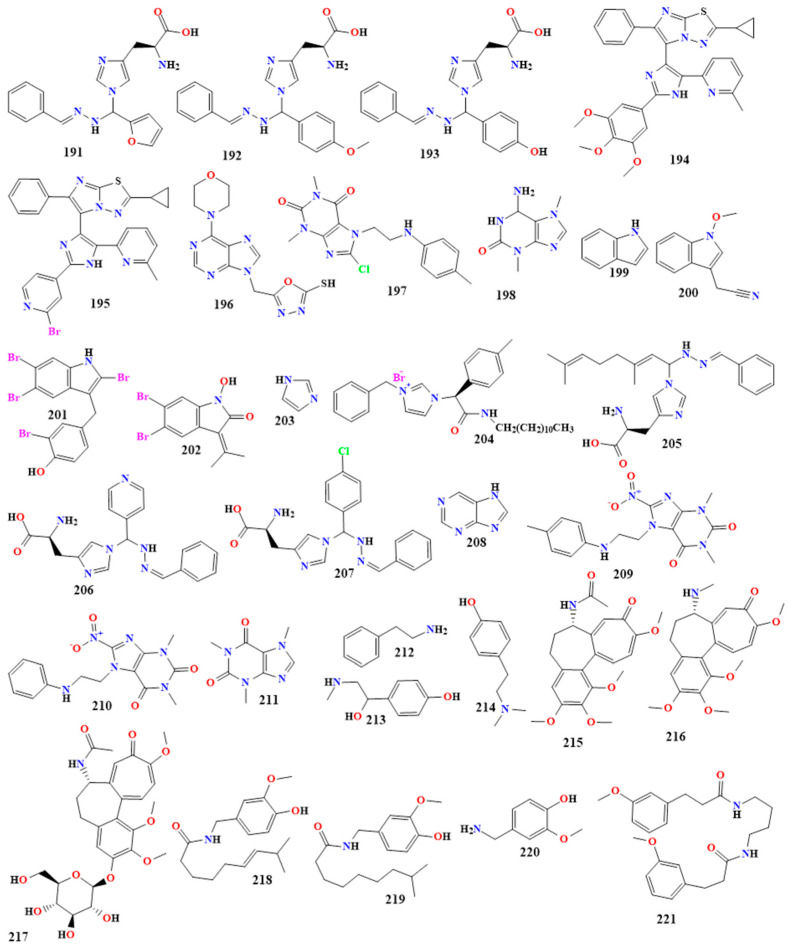
Chemical structures of alkaloids **191**–**221**.

**Figure 12 toxins-16-00489-f012:**
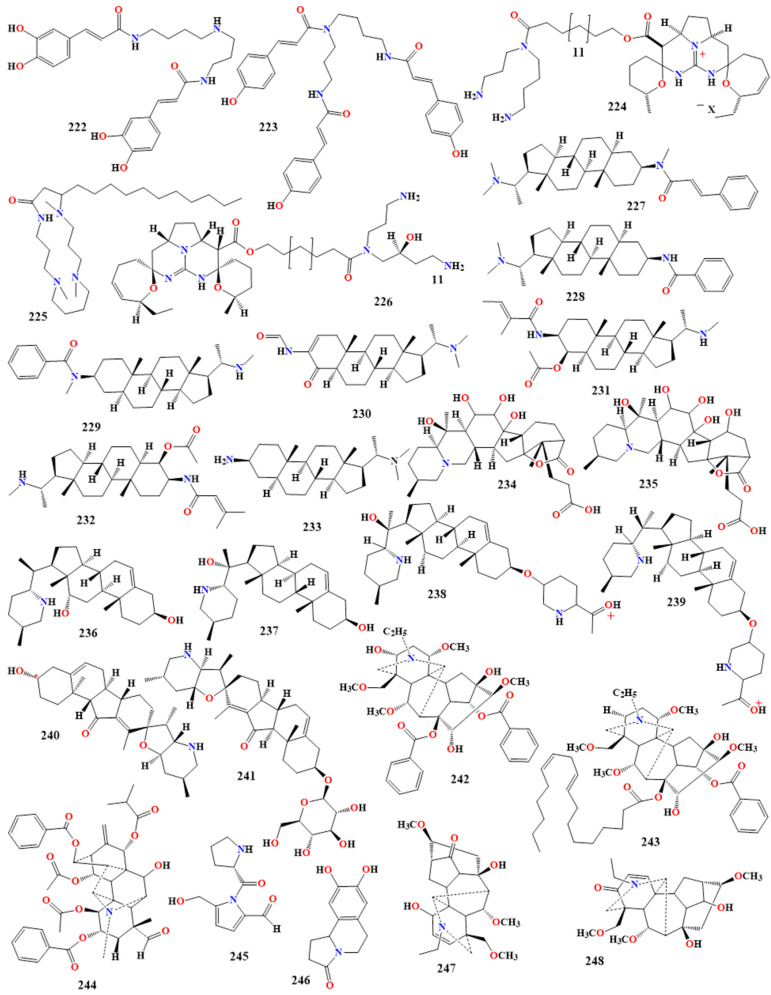
Chemical structures of alkaloids **222**–**248**.

**Table 1 toxins-16-00489-t001:** Antimicrobial true alkaloid compounds **7**–**24**.

Type	Compound	Structure	Source	Antimicrobial Activity
Pyrrolidine	Hygrine [41,42]	**7** in Figure 3	Peruvian coca shrub (*Erythroxylon truxillense* Rusby).	Antifungal, antitubercular, and antibacterial activities.
	Cuscohygrine [42,43]	**8** in Figure 3	Peruvian coca shrub (*Erythroxylon truxillense* Rusby).	Antifungal, antitubercular, and antibacterial activities.
	6-(pyrrolidin-2-yl)DAPG [44]	**9** in Figure 3	*Pseudomonas protegens* UP46.	Antibacterial activity against *S. aureus* and *B. cereus*, with MIC values ranging from 2 to 4 µg/mL.
Tropane	Cocaine [45]	**10** in Figure 3	*Erythroxylum coca.*	Four percent cocaine, demonstrated antibacterial activity against common nasal pathogens.
Pyrrolizidine	Usaramine [46]	**11** in Figure 3	*Crotalaria pallida*.	Antibiofilm activity against *Staphylococcus epidermidis*.
	Senecionine [47]	**12** in Figure 3	*Tussilago farfara* L.–coltsfoot, Asteraceae, leaf.	Antiviral activity.
	Senkirkine [47]	**13** in Figure 3	*Tussilago farfara* L.–coltsfoot, Asteraceae, leaf.	Antiviral activity.
Piperidine	Piperlongumine [48]	**14** in Figure 3	*Piper longum* L.	Antibacterial activity against *K. pneumoniae* and *P. aeruginosa.*
	Epidihydropini-dine [49]	**15** in Figure 3	*Picea abies* (L.) Karsten.	Antibacterial and antifungal activities against *P. aeruginosa*, *E. faecalis*, *Candida glabrata*, *C. albicans*, *Salmonella enterica*, *B. cereus*, and *S. aureus*, with MIC values ranging from 5.37 to 43 µg/mL.
	6-(piperidin-2-yl)DAPG [44]	**16** in Figure 3	*P. protegens* UP46.	Activity against *S. aureus* and *B. cereus*, with an MIC value of 2 µg/mL.
Quinolizidin	Sophoridine [50]	**17** in Figure 3	*Thermopsis lanceolata* R. Brown.	Active against *E. coli*, *Enterobacter aerogenes*, *P. vulgaris*, *B. subtilis*, and *S. epidermidis*, with MIC values ranging from 0.02 to 0.04 M.
	Sophoramine [51]	**18** in Figure 3	*T. lanceolata* R. Brown	Active against *E. coli*, *E. aerogenes*, *Proteus vulgaris*, *B. subtilis*, and *S. epidermidis*, with MIC values ranging from 0.04 to 0.05 M.
	Cytisine [52,53]	**19** in Figure 3	*T.lanceolata* R. Brown	Active against *E. coli*, *E. aerogenes*, *P. vulgaris*, *B. subtilis*, and *S. epidermidis*, with MIC values ranging from 0.03 to 0.05 M.
	Oxymatrine [54,55]	**20** in Figure 3	*T.lanceolata* R. Brown	Active against *E. coli*, *E. aerogenes*, *P. vulgaris*, *B. subtilis*, and *S. epidermidis*, with an MIC value of 0.05 M.
Indolizidine	Tylophorinine [56]	**21** in Figure 4	*Tylophora indica* *Tylophora atrofolliculata*, *Tylophora mollissima*	Antiviral activity against the transmissible gastroenteritis virus.
	Septicine [57,58]	**22** in Figure 4	*Tylophora indica*	Antibacterial activity against *B. subtilis*, *S. aureus*, *M. luteus*, and *P. aeruginosa.* Antifungal activity against *A. niger*, *Aspergillus fumigatus* and *Trichoderma viride.*
	Caranine [59]	**23** in Figure 4	*Amaryllidaceae*	Active against *Candida dubliniensis*, with an MIC value of 128 µg/mL.
	Lycorine [59]	**24** in Figure 4	*Amaryllidaceae*	Active against *C. dubliniensis*, *C. albicans* and *Lodderomyces elongisporus*, with MIC values ranging from 32 to 64 µg/mL.
			*Pancratium Foetidum* Pom.	Active against *S. aureus*, *B. cereus*, *P. aeruginosa* and *Enterobacter cloacae*, with an MIC value of 0.24 mg/mL.

**Table 2 toxins-16-00489-t002:** Antimicrobial true alkaloid compounds **73**–**97**.

Type	Compound	Structure	Source	Antimicrobial Activity
Pyridine	2-methoxy-4-(2-(2-pyridine)ethy)- phenol [104]	**73** in Figure 6	*Zingiberis rhizome.*	Antifungal activity against *C. albicans*, ith a 1.0 mg/mL MIC value.
Isoquinoline	Chelerythrine [105]	**74** in Figure 6	*Toddalia asiatica* (Linn) Lam.	Activity against *S. aureus* and MRSA.
	Palmatine [106]	**75** in Figure 6	*Coptis chinensis*.	Activity against *Helicobacter pylori*, with MIC values ranging from 75 to 200 μg/mL.
	Tetrandrine [107]	**76** in Figure 6	*Stephania tetrandra* S. Moore.	Activity against *S. aureus*, with MIC values ranging from 125 to 250 μg/mL.
	Bicuculline [108]	**77** in Figure 6	*Fumaria and Corydalis.*	Antibacterial activity against *E. coli*, *P. aeruginosa*, *P. mirabilis*, *K. pneumoniae*, *A. baumannii*, *S. aureus* and *B. subtilis*, as well as antifungal activity against *C. albicans.*
	Protopine [109]	**78** in Figure 6	*Fumaria and Corydalis.*	Antibacterial activity against *E. coli*, *P. aeruginosa*, *P. mirabilis*, *K. pneumoniae*, *A. baumannii*, *S. aureus* and *B. subtilis*, as well as antifungal activity against *C. albicans.*
	Magnoflorine [110]	**79** in Figure 6	*Coptidis rhizome*, *Mume fructus*, *Schizandrae fructus*, and *Magnolia grandiflora.*	Antimicrobial activity against enterohemorrhagic *E. coli*, *S. aureus*, and *C. albicans.* It also has antiviral activity against HSV-1.
Oxazole	2-terButyl-4-(4-chlorophenyl)oxa-zole [111]	**80** in Figure 6	Synthesized compounds.	Activity against *B. subtilis*, *S. aureus*, *E. coli*, and *K. pneumonia.*
	4-(4-bromophenyl)-2-tert-butylo-azole [111]	**81** in Figure 6	Synthesized compounds.	Activity against *B. subtilis*, *S. aureus*, *E. coli*, and *K. pneumonia.*
	4-(5-methyl-1,2,4-oxadiazol-3-yl-phenol [112]	**82** in Figure 6	Synthesized compounds.	MIC of 25 μg/mL against *S. aureus* and *A. niger*
	4-(3-methyl-1,2,4-oxadiazol-5-yl-phenol [112]	**83** in Figure 6		MIC of 25 μg/mL against *E. coli* and *A. niger*
	4-(5-methyl-1,3,4-oxadiazol-2-yl-phenol [112]	**84** in Figure 6		MIC of 25 μg/mL against *S. aureus*, *E. coli* and *A. niger*
Isoxazole	(E)-4-((5-amino-4-(phenyldiazen-yl)-1H-pyrazol-3-yl)amino)-N-(5-methylisox zol-3-yl)benzenesulfonamide [113]	**85** in Figure 6	Synthesized compounds.	Antibacterial activity against *S. pneumoniae*, *B. subtilis*, *P. aeruginosa*, and *E. coli*, with zone inhibition diameters of 16.70, 19.20, 13.30 and 13.60 cm, respectively. In addition, antifungal activity against *A. fumigatus*, *Syncephalastrum racemosum*, *G. candidum*, and *C. albicans*, with zone inhibition diameters 16.80, 13.40, 19.60 and 15.90 cm, respectively.
	(E)-4-((5-amino-4-((4-methoxyph-enyl)diazenyl)-1H-pyrazol-3-yl)a-mino)-N-(5-methylisoxazol-3-yl)-benzenesulfonamide [113]	**86** in Figure 6	Synthesized compounds.	Antibacterial activity against *S. pneumoniae*, *B. subtilis*, *P. aeruginosa*, and *E. coli*, with zone inhibition diameters of 18.30, 22.60, 19.30, and 17.80 cm, respectively. In addition, antifungal activity against *A. fumigatus*, *S. racemosum*, *G. candidum*, and *C. albicans*, with zone inhibition diameters of 20.60, 16.70, 22.40, and 17.60 cm, respectively.
	(E)-4-((5-amino-4-((4-chlorophenyl)diazenyl)- 1H-pyrazol-3-yl)- amino)-N-(5-methylisoxazol- -3-yl)benzenesulfonamide [113]	**87** in Figure 7	Synthesized compounds.	Antibacterial activity against *S. pneumoniae*, *B. subtilis*, *P. aeruginosa*, and *E. coli* with zone inhibition diameters of 23.0, 32.40, 17.30, and 19.90 cm, respectively. In addition, antifungal activity against *A. fumigatus*, *S. racemosum*, *G. candidum*, and *C. albicans*, with zone inhibition diameters of 23.70, 19.70, 28.70 and 25.40 cm, respectively.
	4-(6-amino-3,5-dicyano-4-(4-met- hoxyphenyl)-2-oxopyridin- -1(2H)-yl)-N-(5-methylisoxazol-3-yl)benzenesulfonamide [113]	**88** in Figure 7	Synthesized compounds.	Antibacterial activity against *S. pneumoniae*, *B. subtilis*, *P. aeruginosa*, and *E. coli* with zone inhibition diameters of 16.9, 18.2, 9.8 and 11.9 cm, respectively. In addition, antifungal activity against *A. fumigatus*, *S. racemosum*, *G. candidum*, and *C. albicans*, with zone inhibition diameters of 16.2, 15.0, 17.6 and 14.10 cm, respectively.
Thiazole	Camalexin [114]	**89** in Figure 7	*Arabidopsis thaliana.*	Antifungal activity against *Botrytis cinerea*
	(E)-2-(1-(2-(4-butylphenyl)-4-methylthiazol-5-yl)ethylidene)hy-drazinecarboximidamide [115]	**90** in Figure 7	Synthesized compounds.	Exhibited MIC values ranging from 1.38 to 2.77 μg/mL against MRSA, VISA and VRSA
	(E)-2-(1-(2-([1,1′-biphenyl]-4-yl)-4-methylthiazol-5-yl)ethylidene)hydrazinecarboximidamide [115]	**91** in Figure 7	Synthesized compounds.	Exhibited MIC values ranging from 0.70 to 1.40 μg/mL against MRSA, VISA and VRSA
Quinazoline	2-(6,7-Dimethoxy-3- benzyl-4-oxo-3,4-dihydro quinazoline-2-ylthio) nicotinic acid [116]	**92** in Figure 7	Synthesized compounds.	Significant antimicrobial activity.
	1-(4-Hydroxybenzylidene)-4-(4-(4-oxo-2-phenylquinazolin-3(4H)-yl) phenyl) semicarbazide [117]	**93** in Figure 7	Synthesized compounds.	Antibacterial activity against *S. aureus*, *S. epidermidis*, *M. luteus*, *B. cereus*, *E. coli*, and *P. aeruginosa.*
	6-(2-Thienyl)-2-iodo-benzimidaz- o [1,2-c]-quinazoline [118]	**94** in Figure 7	Synthesized compounds.	Antibacterial and antifungal activity against *E. coli*, *S. aureus*, *B. subitilis*, *S. cerevisiae*, and *C. albicans.*
Quinoline	Antidesmone [119]	**95** in Figure 7	*Waltheria indica.*	Activity against *Sclerotinia sclerotiorum*, *Botryosphaeria dothidea*, *Pestalotiopsis guepinii*, *Colletotrichum musae*, *Colletotrichum orbiculare*, *Parasterope longiseta*, and *Phytophthora nicotianae.*
	Waltherione C [119]	**96** in Figure 7	*Waltheria indica.*	Activity against *S. sclerotiorum*, *B. dothidea*, *Pestalotiopsis guepini*, *C. musae*, *C. orbiculare*, *P. longiseta*, and *P. nicotianae.*
	Evocarpine [120]	**97** in Figure 7	Fructus Euodiae.	Significant activity against MRSA, with an MIC value of 8 μg/mL, while the standard drugs oxacillin, erythromycin, and tetracycline had MIC values of ≥128 μg/mL.

**Table 3 toxins-16-00489-t003:** Antimicrobial true alkaloid compounds **171**–**198**.

Type	Compound	Structure	Source	Antimicrobial Activity
Acridine	Shermilamine B [168]	**171** in Figure 10	Ascidian *C. dellechiajei.*	Activity against *E. coli* and *M. luteus*, with MIC values ranging from 2.0 to 8.0 µM.
	*N*-deacetylkuanoniamine D [168]	**172** in Figure 10		Activity against *E. coli* and *M. luteus*, MIC values ranging from 2.5 to 5 µM.
	*N*-deacetylshermilamine B [168]	**173** in Figure 10		Activity against *E. coli* and *M. luteus*, MIC values ranging from 1.1 to 4.5 µM.
	11-hydroxyascididemin [168]	**174** in Figure 10		Activity against *E. coli* and *M. luteus*, MIC values ranging from 2.6 to 10.5 µM.
	Cystodimine A [171]	**175** in Figure 10		Activity against *E. coli* and *M. luteus*, MIC values ranging from 1.2 to 2.4 µM.
	Cystodimine B [171]	**176** in Figure 10		Activity against *E. coli* and *M. luteus*, MIC values ranging from 2.6 to 10.5 µM.
	Ascididemin [171]	**177** in Figure 10		Activity against *E. coli* and *M. luteus*, MIC values ranging from 0.2 to 0.3 µM.
Indole	Latifolianine A [172]	**178** in Figure 10	*Nauclea latifolia.*	Activity against *Haemophilus influenzae*, with an MIC value of 25 µg/mL.
	Latifoliaindole A [172]	**179** in Figure 10		Activity against *H. influenzae*, with an MIC value of 50 µg/mL.
	Latifoliaindole B [172]	**180** in Figure 10		Activity against *H. influenzae*, with an MIC value of 25 µg/mL.
	Alstoniascholarine A [173]	**181** in Figure 10	*Alstonia scholaris.*	*Activity against P. aeruginosa*, *K. pneumoniae*, *E. coli*, and *E. faecalis*, with MIC values ranging from 25 to 50 μg/mL.
	Alstoniascholarine C [173]	**182** in Figure 10		*Activity against P. aeruginosa*, *K. pneumoniae*, *E. coli*, and *E. faecalis*, with MIC values ranging from 12.5 to 50 μg/mL.
	Alstoniascholarine E [173]	**183** in Figure 10		*Activity against P. aeruginosa*, *K. pneumoniae*, *E. coli*, and *E. faecalis*, with MIC values ranging from 25 to 50 μg/mL.
	Alstoniascholarine F [173]	**184** in Figure 10		*Activity against P. aeruginosa*, *K. pneumoniae*, *E. coli*, and *E. faecalis*, with MIC values ranging from 3.13 to 50 μg/mL.
	Alstoniascholarine H [173]	**185** in Figure 10		*Activity against P. aeruginosa*, *K. pneumoniae*, and *E. coli*, with MIC values ranging from 25 to 50 μg/mL.
	Alstoniascholarine I [173]	**186** in Figure 10		*Activity against P. aeruginosa*, *K. pneumoniae*, and *E. coli*, with MIC values ranging from 12.5 to 25 μg/mL.
	Alstoniascholarine J [173]	**187** in Figure 10		*Activity against S. aureus*, *P. aeruginosa*, *E. faecalis*, *K. pneumoniae* and *E. coli*, MIC values ranging from 3.13 to 25 μg/mL.
	Scholarisine T [174]	**188** in Figure 10		Activity against *E. coli*, *B. subtilis* and *S. typhi*, MIC values ranging from 0.78 to 12.5 μg/mL.
	Scholarisine U [174]	**189** in Figure 10		Activity against *E. coli* and *B. subtilis*, MIC values ranging from 0.78 to 3.12 μg/mL.
	Scholarisine V [174]	**190** in Figure 10		Activity against *E. coli*, *S. aureus* and *S. typhi*, MIC values ranging from 0.78 to 12.5 μg/mL.
Imidazole	(2S)-2-Amino-3-(1-(((E)-2-benzylidenehydrazinyl)-(furan-2-yl)methyl)-1H-imidazol-4-yl)propanoic acid [175]	**191** in Figure 11	Synthesized compound.	Antibacterial activity against *S. aureus* and *K. pneumoniae*, with an MIC value of 0.5 μg/mL.
	(2S)-2-Amino-3-(1-(((E)-2-benzylidenehydrazinyl)-(4-methoxyphenyl)methyl)-1H-imidazol-4-yl)propanoic acid [175]	**192** in Figure 11	Synthesized compound.	Antibacterial activity against *K. pneumoniae*, with an MIC value of 32 μg/mL.
	(2S)-2-Amino-3-(1-(((E)-2-benzylidenehydrazinyl)-(4-hydroxyphenyl)methyl)-1H-imidazol-4-yl)propanoic acid [175]	**193** in Figure 11	Synthesized compound.	Antifungal activity against *C. albicans*, with an MIC value of 0.5 μg/mL. In addition, it possessed remarkable activity against *A. niger*, with an MIC value of 16 μg/mL, while the standard drug clotrimazole had an MIC value of 32 μg/mL.
	2-cyclopropyl-5-(5-(6-methylpyridin-2-yl)-2-(3,4,5-trimethoxyphenyl)-1H-imidazol-4-yl)-6-phenylimidazo[2,1-b][1,3,4]thiadiazole [176]	**194** in Figure 11	Synthesized compound.	Antibacterial activity, with MIC values ranging from 1 to 2 μg/mL.
	5-(2-(2-bromopyridin-4-yl)-5-(6-methylpyridin-2-yl)-1H-imidazol-4-yl)-2-cyclopropyl-6-phenylimidazo[2,1-b][1,3,4] thiadiazole [176]	**195** in Figure 11	Synthesized compound.	Antibacterial activity, with an MIC value of 0.5 μg/mL.
Purine	5-((6-Morpholino-9H-purin-9-yl)methyl)-1,3,4-oxadiazole-2-Thiol [177]	**196** in Figure 11	Synthesized compound.	Demonstrated strong inhibitory effect against *Xanthomonas oryzae*, with an EC50 value of 8.39 μg/mL.
	8-chloro-1,3-dimethyl-7-{2-[(4- methylphenyl)amino] ethyl}-3,7-dihydro-1H-purine-2,6-dione [178]	**197** in Figure 11	Synthesized compound.	Antibacterial activity against *E. coli*, *P. aeruginosa*, *B. subtilis*, and *S. aureus.*
	3,7-Dimethylisoguanine [179]	**198** in Figure 11	Sponge *A. dilatate*	Activity against *K. pneumoniae* and *P. aeruginosa*

**Table 4 toxins-16-00489-t004:** Alkaloids that are most antimicrobial.

Compound No.	Compound Name
**91**	(*E*)-2-(1-(2-([1,1′-biphenyl]-4-yl)-4-methylthiazol-5-yl)ethylidene)hydrazinecarboximidamide
**124**	2-(Cyclohexylamino)-1-(5-nitrothiophen-2-yl)-2-oxoethyl-5-amino-3-methyl-1,2-oxazole-4-carboxylate
**125**	2-(benzylamino)-1-(5-nitrothiophen-2-yl)- 2-oxoethyl-5-amino-3-methyl-1,2-oxazole-4-carboxylate
**136**	7-Methyl-3-(4-methyl-2-(methylamino)thiazol-5-yl)-5*H*-thiazolo[3,2-*a*]pyrimidin-5-one
**137**	2-(2-(1-(4-Bromophenyl)ethylidene)hydrazineyl)-*N*,4′-dimethyl-[4,5′-bithiazol]-2′-amine
**138**	2-(2-(1-(4-(4-Chlorophenyl)-5-imino-4,5-dihydro-1,3,4-thiadiazol-2-yl)ethylidene)-hydrazineyl)-*N*,4′-dimethyl-[4,5′-bithiazol]-2′-amine
**163**	2-{2-[(2-Chloroquinolin-3-yl)methylene]hydrazinyl} acetonitrile
**164**	6-(2-Chloroquinolin-3-yl)-4-(4-aminophenyl) pyrimidin-2(1H )-one
**191**	(2S)-2-Amino-3-(1-(((E)-2-benzylidenehydrazinyl)(furan-2-yl)methyl)-1H-imidazol-4-yl)propanoic acid
**193**	(2S)-2-Amino-3-(1-(((E)-2-benzylidenehydrazinyl)-(4-hydroxyphenyl)methyl)-1H-imidazol-4-yl)propanoic acid
**205**	(2S)-2-Amino-3-(1-((E)-1-((E)-2-benzylidenehydrazinyl)-3,7-dimethylocta-2,6-dien-1-yl)-1H-imidazol-4-yl)propanoic acid
**206**	(2S)-2-Amino-3-(1-(((E)-2-benzylidenehydrazinyl)(pyridin-4-yl)methyl)-1H-imidazol-4-yl)propanoic acid
**207**	(2S)-2-Amino-3-(1-(((E)-2-benzylidenehydrazinyl)(4-chlorophenyl)methyl)-1H-imidazol-4-yl)propanoic acid
**224**	Ptilomycalin A

## Data Availability

No new data were created or analyzed in this study. Data sharing is not applicable to this article.

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
