# Peer review of "Antibacterial Activity and Antifungal Activity of Monomeric Alkaloids"

_toxins, 2024, doi:10.3390/toxins16110489_

Round 1
Reviewer 1 Report
Comments and Suggestions for Authors
This review covered the topic in some detail and is of interest. This reviewer is in favor of its publication in Toxins.
Author Response
R: This review covered the topic in some detail and is of interest. This reviewer is in favor of its publication in Toxins.
A: Thank you very much for your kind interest.
NOTE: The title was changed to “Antibacterial Activity and Antifungal Activity of Monomeric Alkaloids” by comments from other reviewers.
Reviewer 2 Report
Comments and Suggestions for Authors
Dear authors,
The review article aims to investigate the antibacterial and antifungal properties of natural or synthesized alkaloids and their mechanisms of action against pathogens. Although the authors worked hard to supply the data, I believe this review requires considerable revisions. For this, I have a few remarks and recommendations that must be considered to render this manuscript suitable for publishing.
1. Abstract
The abstract should be restructured and rewritten.
a) The authors can introduce alkaloids and their correlation to antibacterial and antifungal properties.
b) The authors should address the search methodology, selection criteria, and timeframe of their literature review.
c) The authors should mention some of the most potent alkaloids, their specific targets, and their mechanism of action.
2. Introduction
This section should also be rewritten and can encompass the following information.
a) The authors can elucidate how phytoanticipins and phytoalexins primarily regulate bacterial and fungal activity.
b) The authors can provide the fundamental structures of several alkaloids and identify which categories possess significant antibacterial and antifungal properties (may elaborate under the ‘Alkaloids classification’ sub-section).
c) It is advisable to demonstrate the differential effects of alkaloids on Gram-positive and Gram-negative bacteria, in general.
3. Methodology
The authors claimed that 248 alkaloids had been evaluated for antibacterial and antifungal properties. I suggest the authors include a methodology section to address the following questions.
a) How can the authors ensure they have not overlooked any additional alkaloids with antibacterial or antifungal activity?
b) By what criteria did the authors select the relevant literature?
c) What publication timeline did they consider?
d) What were the inclusion and exclusion criteria?
4. Other Suggestions
a) For manuscripts of this nature, it is advisable to discuss the structure-activity relationship.
b) The English language and style require revision throughout the manuscript.
Comments on the Quality of English Languagecan be improved.
Author Response
R: The review article aims to investigate the antibacterial and antifungal properties of natural or synthesized alkaloids and their mechanisms of action against pathogens. Although the authors worked hard to supply the data, I believe this review requires considerable revisions. For this, I have a few remarks and recommendations that must be considered to render this manuscript suitable for publishing.
A: First, I would like to thank you for your time reviewing the manuscript. Your suggestions were much appreciated and improved the work meaningfully and originally. I am grateful for all your hard work.
- Abstract
R: The abstract should be restructured and rewritten.
- a) The authors can introduce alkaloids and their correlation to antibacterial and antifungal properties.
- b) The authors should address the search methodology, selection criteria, and timeframe of their literature review.
- c) The authors should mention some of the most potent alkaloids, their specific targets, and their mechanism of action.
A: The abstract has been modified, and your valuable comments have been taken into consideration.
- Introduction
R: This section should also be rewritten and can encompass the following information.
- a) The authors can elucidate how phytoanticipins and phytoalexins primarily regulate bacterial and fungal activity.
A: There are several forms of classification of alkaloids; the classification is based on the chemical structure of the compounds in this review, as mentioned in the abstract, which has been modified based on your recommendations. For the classification of alkaloids into phytoanticipins and phytoalexins, the mechanisms of antimicrobial action are explained in Section 4, and some mechanisms are mentioned when the compound is mentioned.
- b) The authors can provide the fundamental structures of several alkaloids and identify which categories possess significant antibacterial and antifungal properties (may elaborate under the ‘Alkaloids classification’ sub-section).
A: The classification was shown in the fifth section, where the basic structure of each class was drawn separately in the figures, along with the compounds derived from it.
- c) It is advisable to demonstrate the differential effects of alkaloids on Gram-positive and Gram-negative bacteria, in general.
A: Each compound mentioned throughout the text from subsection 5.1.1 to 5.4.2 is given its activity against bacteria, both positive and negative, and its activity against fungi, based on clinical studies of the compound.
- Methodology
R: The authors claimed that 248 alkaloids had been evaluated for antibacterial and antifungal properties. I suggest the authors include a methodology section to address the following questions.
- a) How can the authors ensure they have not overlooked any additional alkaloids with antibacterial or antifungal activity?
- b) By what criteria did the authors select the relevant literature?
- c) What publication timeline did they consider?
- d) What were the inclusion and exclusion criteria?
A: As you recommended, all answers to the queries are presented in the new revised abstract and introduction.
- Other Suggestions
R: a) For manuscripts of this nature, it is advisable to discuss the structure-activity relationship.
A: Added: in pyrrolidine alkaloids, piperidine derivatives, pyridine derivatives, oxazole derivatives, isoxazole derivatives, thiazole derivatives, indole derivatives sections in orange color font
R: b) The English language and style require revision throughout the manuscript.
A: Done
NOTE: The title was changed to “Antibacterial Activity and Antifungal Activity of Monomeric Alkaloids” by comments from other reviewers.
Reviewer 3 Report
Comments and Suggestions for Authors
The Authors submitted a thorough review of all kinds of alkaloids with antimicrobial activity and reported in the last ten years. I found the work very compelling and very clearly structured, with all topics (classification, modes of action, …) nicely introduced. The main body of the review (Point 5), however, is improvable. The text is easy to read, although I found it a bit tedious and repetitive. All compounds are named the same way, describing the structure and referring to the corresponding figure and table, and briefly discussing their activity. I think that a bit of further discussion on which compounds are the most promising ones, and if any of them is candidate to clinical trials (or already in them) should be incorporated.
I would modify the format of the figures, of most of them. Figures are too condensed, depicting too many molecular structures in too little space (e.g. twenty structures in Figure 3, just occupying half page; thirty-one structures in Figure 11). I would reconfigure them with more space between structures.
The sentence in lines 172-174 (“The gram-negative bacteria were mostly active against K. Pneumonia, P. aeruginosa, Shigella boydii, and Salmonella typhi, with a minor activity observed against E. coli.”) must be rephrased (gram negative bacteria are not active against P. aeruginosa and other microorganisms, the alkaloid is).
Some of the numerals by which compounds are referred to are not in bold characters (e.g. “41” in line 256, “43” in line 268, and few more throughout the text). Please amend.
In line 205 there are two spaces too many (say “9-(2,3-dihydroxybutyryl)” and “7-(2-methylbutyryl)”).
In line 223, there is no need of capital “D” to name two compounds (“2,6-dipiperidino-4-bromochlorobenzene and 2,6-dipiperidino-4…”), as it is not the beginning of a sentence. The same applies to lines 443 (“itraconazole”), 453 (“… 2-(pyrrolidin-1-ylmethyl)”…) and 454 (“…3-[(4-nitrobenzylidene)…”), 480, 481, …
In line 226, I would say “… (25-30, Figure 4) were investigated of their …” (or rephrase the sentence, as “The antimicrobial activity of six (…) was investigated.”).
In line 227, “most minor” is not correct, replace it by “lowest”.
In line 251, should it be “cermicines C” or “cermicine C”? The same in line 253 (“jussiaeiines B” or “jussiaeiine B”?).
In table 1, 3 rd column, I would not use hyphenation (as in “Fig-ure”). I would perhaps abbreviate it (“Fig.”) or reformat the table. The same is seen in Table 3.
The description given in line 283 on the structure of pyridine is not formally correct. It is not an unsaturated nitrogen; it is the entire six-membered heterocycle which is unsaturated.
Delete “(65 and 66 in Figure 6)” in line 292, it is already said one line before.
In lines 349-350, the Authors say: “In one investigation, researchers developed a novel family of antibacterial drugs called pyrazoles, including 2,4-disubstituted oxazol-5-one (3a-3g).”. Pyrazoles and oxazoles (or oxazol-5-ones) are not the same structural motifs. I think that the name “pyrazoles” does not apply to these structures.
In lines 407 and 411, use subindexes for “4-CH3” and “4-OCH3”. I suggest revising for other of those small mistakes.
Some words (names of compounds) in table 2, 2nd column should be hyphenated (or the table may be reformatted, improving the widths of the columns or reducing the sizes of the fonts).
In line 503, there should be no comma (“The acridine derivative 1-phenyl-…”). Besides, the name given to structure 168 in Figure 10 (“1-phenyl-4-(7-benz[c]acridinyl) thiosemicarbazide”) is not correct. It should be “1-benzoyl” instead of “1-phenyl”.
In line 562, I would say “fused” instead of “connected”.
In line 610, “Synephrine (213 in Figure 11)” is written in bold characters.
In line 620, two decimals are enough (“9.77” instead of “9.767”).
I found Point 6 (“Future Implications for Healthcare”) rather weak. The arguably main problem of antimicrobials (resistant strains) is just very briefly stated, and I think that some lines on ongoing attempts to overcome bacteria and fungi resistance to antimicrobials must be added, also with some literature references (there is none in Point 6).
In pages 36 and 37, the titles of the works in references [118] and [123] are given in capital letters. Those should be written in small case in agreement with the other literature citations.
I recommend major modifications.
Author Response
R: The Authors submitted a thorough review of all kinds of alkaloids with antimicrobial activity and reported in the last ten years. I found the work very compelling and very clearly structured, with all topics (classification, modes of action, …) nicely introduced. The main body of the review (Point 5), however, is improvable. The text is easy to read, although I found it a bit tedious and repetitive. All compounds are named the same way, describing the structure and referring to the corresponding figure and table, and briefly discussing their activity. I think that a bit of further discussion on which compounds are the most promising ones, and if any of them is candidate to clinical trials (or already in them) should be incorporated.
A: First, I would like to thank you for your time in reviewing the manuscript. Your suggestions were much appreciated and improved the work meaningfully and originally. I am glad about your hard work.
R: I would modify the format of the figures, of most of them. Figures are too condensed, depicting too many molecular structures in too little space (e.g. twenty structures in Figure 3, just occupying half page; thirty-one structures in Figure 11). I would reconfigure them with more space between structures.
A: Thank you for your recommendation. The journal ultimately determines the figures' style and the components' distribution to conform to its publishing specifications and conditions. Usually, the MDPI group contacts the author to confirm changes to the order of figures before publication.
R: The sentence in lines 172-174 (“The gram-negative bacteria were mostly active against K. Pneumonia, P. aeruginosa, Shigella boydii, and Salmonella typhi, with a minor activity observed against E. coli.”) must be rephrased (gram negative bacteria are not active against P. aeruginosa and other microorganisms, the alkaloid is).
A: Modified ((Red))
R: Some of the numerals by which compounds are referred to are not in bold characters (e.g. “41” in line 256, “43” in line 268, and few more throughout the text). Please amend.
A: Done (Blue)
R: In line 205 there are two spaces too many (say “9-(2,3-dihydroxybutyryl)” and “7-(2-methylbutyryl)”).
A: Done (yellow)
R: In line 223, there is no need of capital “D” to name two compounds (“2,6-dipiperidino-4-bromochlorobenzene and 2,6-dipiperidino-4…”), as it is not the beginning of a sentence. The same applies to lines 443 (“itraconazole”), 453 (“… 2-(pyrrolidin-1-ylmethyl)”…) and 454 (“…3-[(4-nitrobenzylidene)…”), 480, 481
A: Done (yellow)
R: In line 226, I would say “… (25-30, Figure 4) were investigated of their …” (or rephrase the sentence, as “The antimicrobial activity of six (…) was investigated.”).
A: Done (Pink)
R: In line 227, “most minor” is not correct, replace it by “lowest”.
A: Done (Pink)
R: In line 251, should it be “cermicines C” or “cermicine C”? The same in line 253 (“jussiaeiines B” or “jussiaeiine B”?).
A: Done (Gold)
R: In table 1, 3 rd column, I would not use hyphenation (as in “Fig-ure”). I would perhaps abbreviate it (“Fig.”) or reformat the table. The same is seen in Table 3.
A: Done in Table 1 and Table 3
R: The description given in line 283 on the structure of pyridine is not formally correct. It is not an unsaturated nitrogen; it is the entire six-membered heterocycle which is unsaturated.
A: Modified (green)
R: Delete “(65 and 66 in Figure 6)” in line 292, it is already said one line before.
A: Deleted; thank you very much.
R: In lines 349-350, the Authors say: “In one investigation, researchers developed a novel family of antibacterial drugs called pyrazoles, including 2,4-disubstituted oxazol-5-one (3a-3g).”. Pyrazoles and oxazoles (or oxazol-5-ones) are not the same structural motifs. I think that the name “pyrazoles” does not apply to these structures.
A: The sentence was modified to read, "In one investigation, researchers developed a novel family of antibacterial drugs including 2,4-disubstituted oxazol-5-one (3a-3g) .......” where the group name was deleted, pyrazoles.
R: In lines 407 and 411, use subindexes for “4-CH3” and “4-OCH3”. I suggest revising for other of those small mistakes.
A: Modified (green)
R: Some words (names of compounds) in table 2, 2nd column should be hyphenated (or the table may be reformatted, improving the widths of the columns or reducing the sizes of the fonts).
A: As the journal has determined, we cannot change or reduce table font size. However, the column width has been modified.
R: In line 503, there should be no comma (“The acridine derivative 1-phenyl-…”). Besides, the name given to structure 168 in Figure 10 (“1-phenyl-4-(7-benz[c]acridinyl) thiosemicarbazide”) is not correct. It should be “1-benzoyl” instead of “1-phenyl”.
A: The comma has been removed, and the compound name has been modified (Gold)
R: In line 562, I would say “fused” instead of “connected”.
A: Modified (Gold)
R: In line 610, “Synephrine (213 in Figure 11)” is written in bold characters.
A: Modified (Gold)
R: In line 620, two decimals are enough (“9.77” instead of “9.767”).
A: Modified (Gold)
R: I found Point 6 (“Future Implications for Healthcare”) rather weak. The arguably main problem of antimicrobials (resistant strains) is just very briefly stated, and I think that some lines on ongoing attempts to overcome bacteria and fungi resistance to antimicrobials must be added, also with some literature references (there is none in Point 6).
A: It has been rewritten (brown font)
R: In pages 36 and 37, the titles of the works in references [118] and [123] are given in capital letters. Those should be written in small case in agreement with the other literature citations.
A: Modified and checked (grey)
NOTE: Per other reviewers' comments, the title was changed to Antibacterial Activity and Antifungal Activity of Monomeric Alkaloids.
Reviewer 4 Report
Comments and Suggestions for Authors
The comprehensive manuscript provides an in-depth examination of antibacterial and antifungal alkaloids over the past decade (2014-2024), reviewing a vast array of literature to provide a nuanced understanding of the topic. Following points need to be addressed:
1. Most of the articles that lie under the scope of current topic are missing for example;
https://doi.org/10.1002/cjoc.202000736
https://doi.org/10.1016/j.fitote.2018.06.017
https://doi.org/10.1016/j.phytol.2013.11.016
https://doi.org/10.1038/s41429-020-0333-2
https://doi.org/10.3390/molecules26051375
The authors must cite all corresponding literature within the selected year range.
2. Revise the abstract as the current one fails to provide a brief overview of the topic in real essence. Further the main findings of the literature study should be provided in the concise form.
3. As the authors have mentioned that the reported naturally occurring alkaloids have been synthesized in the past decade, briefly discuss about the adopted synthetic routes.
4. The title of the manuscript must be revised in accordance with content provided in the manuscript.
5. Ensure uniform formatting for repeated terms and phrases (e.g., use term antimicrobial or anti-microbial and apply it throughout).
6. Bold the structure numbering throughout the manuscript.
7. The manuscript must be critically revised as multiple sentences fail to convey clear meaning. The manuscript is loaded with several grammatical mistakes. Which require grave revision.
-Revise the lines 152 to 158 with respect to the conciseness and clarity of the sentence.
-Line 162: Correct the spelling.
-In line 798: Provide the abbreviation for “Mycobacterium tuberculosis” according to the format of the line.
-Line 824 to 829: The line is inappropriate to convey the intended meaning. Alter this line for intended meaning.
-Line 76 to 78: correct the sentence structure.
-Line 48 to 49: required proper formatting.
-Reference 63 does not coincide with line 236.
-Structure 176 is incorrect, also check all the structures from reference paper.
-Line 610: correct the formatting.
Author Response
R: The comprehensive manuscript provides an in-depth examination of antibacterial and antifungal alkaloids over the past decade (2014-2024), reviewing a vast array of literature to provide a nuanced understanding of the topic. Following points need to be addressed:
A: First, I would like to thank you for your time reviewing the manuscript. Your suggestions were much appreciated and improved the work meaningfully and originally. I appreciate all of your hard work.
R: 1. Most of the articles that lie under the scope of current topic are missing for example;
https://doi.org/10.1002/cjoc.202000736
https://doi.org/10.1016/j.fitote.2018.06.017
https://doi.org/10.1016/j.phytol.2013.11.016
https://doi.org/10.1038/s41429-020-0333-2
https://doi.org/10.3390/molecules26051375
R: The authors must cite all corresponding literature within the selected year range.
A: Thank you for the suggestion, but as mentioned, the alkaloids and studies were randomly selected in this study between 2014 and 2024, and the goal was not to cover all alkaloids.
R: 2. Revise the abstract as the current one fails to provide a brief overview of the topic in real essence. Further the main findings of the literature study should be provided in the concise form.
A: Modified (Green font)
R: 3. As the authors have mentioned that the reported naturally occurring alkaloids have been synthesized in the past decade, briefly discuss about the adopted synthetic routes.
A: Synthesis procedures were added and highlighted with blue font.
R: 4. The title of the manuscript must be revised in accordance with content provided in the manuscript.
A: Changed to Antibacterial activity and Antifungal activity of Monomeric Alkaloids
R: 5. Ensure uniform formatting for repeated terms and phrases (e.g., use term antimicrobial or anti-microbial and apply it throughout).
A: Changed all to antimicrobial
R: 6. Bold the structure numbering throughout the manuscript.(Done (Blue))
R: 7. The manuscript must be critically revised as multiple sentences fail to convey clear meaning. The manuscript is loaded with several grammatical mistakes. Which require grave revision.
A: Done
R: Revise the lines 152 to 158 with respect to the conciseness and clarity of the sentence.
A: Done Violet font
R: Line 162: Correct the spelling.
A: "characterized by an N-methyl group" replaced by "characterized by N-methyl group"
R: n line 798: Provide the abbreviation for “Mycobacterium tuberculosis” according to the format of the line.
A: Section 5 has been rewritten (brown font)
R: Line 824 to 829: The line is inappropriate to convey the intended meaning. Alter this line for intended meaning.
A: Modified (Pink)
R: Line 76 to 78: correct the sentence structure.
A: Modified (yellow)
R: Line 48 to 49: required proper formatting.
A: Done, (Red)
R: Reference 63 does not coincide with line 236.
A: Sorry, it's 65 (Red)
R: Structure 176 is incorrect, also check all the structures from reference paper.
A: Corrected
R: Line 610: correct the formatting.
A: corrected (gold)
NOTE: Per other reviewers' comments, the title was changed to Antibacterial Activity and Antifungal Activity of Monomeric Alkaloids.
Round 2
Reviewer 2 Report
Comments and Suggestions for Authors
The revised manuscript looks good. Still, I have a few suggestions.
1. I suggest the authors add concluding remarks in the 'abstract'.
2. Replace the numbering of the metabolites with their chemical name in the 'abstract'.
3. I suggest the authors mention which alkaloid group(s) possess(es) significant antibacterial and antifungal properties.
Author Response
The revised manuscript looks good. Still, I have a few suggestions.
We thank the reviewer for his helpful suggestions.
Comment 1: I suggest the authors add concluding remarks in the 'abstract'.
Response 1: In blue, we added a short concluding remark as the abstract's last sentence: "Given the rise in antibiotic resistance, these alkaloids are highly significant for their potential to create novel antimicrobial drugs”.
Comment 2. Replace the numbering of the metabolites with their chemical name in the 'abstract'.
Response 2: In yellow: Replacing the numbers with the chemical names of the compounds cited in the abstract would make it difficult to read. We have preferred to include a list with the chemical names of the compounds cited in the abstract at the end of Chapter 5.
Comment 3: I suggest the authors mention which alkaloid group(s) possess(es) significant antibacterial and antifungal properties.
Response 3: In blue: At the end of chapter 5, we added the sentence: “As a conclusion to this chapter, we can state that, based on the research, isoquinoline alkaloids and indole alkaloids stand out as being very promising for use in medicine because of their potent activities against harmful microorganisms”.
Reviewer 3 Report
Comments and Suggestions for Authors
The Authors promptly and duly responded to all my suggestions. I can see that the revised version of their manuscript is improved and can be essentially accepted for publication. I only found very few minor mistakes.
In line 153 of the revised version, remove “While”.
In line 177, say “hydroxy” (no need of capital “H”).
In line 308, I would say “…heterocyclic compounds consisting of…”.
In line 330, is it “where” or “whether”?
In line 890, I would say “therapies” instead of “therapy”.
In line 892, I would say “combat” instead of “combating”.
I recommend minor modifications (acceptance once these corrections have been made).
Author Response
We thank the reviewer for his fruitful comments.
- In line 153 of the revised version, remove “While”.
DONE
- In line 177, say “hydroxy” (no need of capital “H”).
DONE
- In line 308, I would say “…heterocyclic compounds consisting of…”.
DONE
- In line 330, is it “where” or “whether”?
Whether
- In line 890, I would say “therapies” instead of “therapy”.
DONE
- In line 892, I would say “combat” instead of “combating”.
DONE
Reviewer 4 Report
Comments and Suggestions for Authors
The authors have significantly revised the manuscript and can now be accepted
Author Response
Thank you very much for your kind evaluation.